# VIDEO SCENE GRAPH GENERATION FROM SINGLE-FRAME WEAK SUPERVISION

**Siqi Chen$^\heartsuit$, Jun Xiao$^\heartsuit$, Long Chen$^\spadesuit$***
$^\heartsuit$Zhejiang University, $^\spadesuit$The Hong Kong University of Science and Technology
{siqic, junx}@zju.edu.cn, zjuchenlong@gmail.com

## ABSTRACT

Video scene graph generation (VidSGG) aims to generate a sequence of graph-structure representations for the given video. However, all existing VidSGG methods are *fully-supervised*, *i.e.*, they need dense and costly manual annotations. In this paper, we propose the first *weakly-supervised* VidSGG task with only single-frame weak supervision: SF-VidSGG. By "weakly-supervised", we mean that SF-VidSGG relaxes the training supervision from two different levels: 1) It only provides single-frame annotations instead of all-frame annotations. 2) The single-frame ground-truth annotation is still a weak image SGG annotation, *i.e.*, an un-localized scene graph. To solve this new task, we also propose a novel Pseudo Label Assignment based method, dubbed as **PLA**. PLA is a two-stage method, which generates pseudo visual relation annotations for the given video at the first stage, and then trains a fully-supervised VidSGG model with these pseudo labels. Specifically, PLA consists of three modules: an object PLA module, a predicate PLA module, and a future predicate prediction (FPP) module. Firstly, in the object PLA, we localize all objects for every frame. Then, in the predicate PLA, we design two different teachers to assign pseudo predicate labels. Lastly, in the FPP module, we fusion these two predicate pseudo labels by the regularity of relation transition in videos. Extensive ablations and results on the benchmark Action Genome have demonstrated the effectiveness of our PLA[1].

## 1 INTRODUCTION

Scene graph (Johnson et al., 2015) is a type of visually-aligned graph-structured representation that summarizes all the object instances (*e.g.*, "`person`", "`chair`") as nodes and their pairwise visual relations (or predicates, *e.g.*, "`sitting on`") as directed edges. As a bridge to connect the vision and language modalities, scene graphs have been widely used in many different downstream visual-language tasks, such as visual captioning (Yang et al., 2019; 2020), grounding (Jing et al., 2020), question answering (Hudson & Manning, 2019), and retrieval (Johnson et al., 2015).

Early Scene Graph Generation (SGG) work mainly focuses on generating scene graphs for the given *image*, dubbed as **ImgSGG** (Xu et al., 2017; Zellers et al., 2018; Chen et al., 2019). However, due to its static nature, ImgSGG fails to represent numerous dynamic visual relations that take place over a period of time, such as "`walking with`" and "`running away`" (vs. static relation "`standing`"). Meanwhile, it is hard or impossible to identify these dynamic visual relations with only a single frame, because these visual relations can only be well classified by considering the temporal context. Therefore, another more meaningful but challenging video-based SGG task was proposed: **VidSGG** (Shang et al., 2017; 2019).

Since the complex and dense annotations of a scene graph (*cf.*, Figure 1(a)), fully-supervised SGG methods always require lots of manual annotations, and the case is even worse for video data. Meanwhile, several prior SGG works (Li et al., 2022a) have found that even carefully manually-annotated labels are still quite noisy, *i.e.*, the annotated positive labels may not be the best matched, and numerous negative labels are just missing annotated. Thus, a surge of recent ImgSGG work (Zareian

---

*Long Chen is the corresponding author.
[1]Codes are available at: https://github.com/zjucsq/PLA.

et al., 2020; Zhong et al., 2021; Shi et al., 2021; Li et al., 2022c) start to generate scene graphs for images with only weak supervision. By "weak supervision", we mean that the annotations for model training are not complete localized scene graphs. For example, a typical type of weak supervision is *unlocalized scene graphs*. As illustrated in Figure 1(b), an unlocalized scene graph only contains image-level object and relation categories without corresponding object bounding boxes (bboxes).

Although recent weakly-supervised ImgSGG has achieved good performance and received unprecedented attention, to the best our knowledge, there is no existing work about generating video scene graphs from weak supervision. To put forward the research on this critical topic, we propose the first weakly-supervised VidSGG task with single-frame weak supervision, called **SF-VidSGG**. Specifically, given an input video, SF-VidSGG aims to generate a localized scene graph for each frame in the video, but *the only supervision for training is an unlocalized scene graph for the middle frame of each training video*. As the example shown in Figure 1(c), the supervision is an unlocalized scene graph for the third frame. Obviously, SF-VidSGG task tries to relieve the intensive annotation issues from two levels: 1) Video-level:

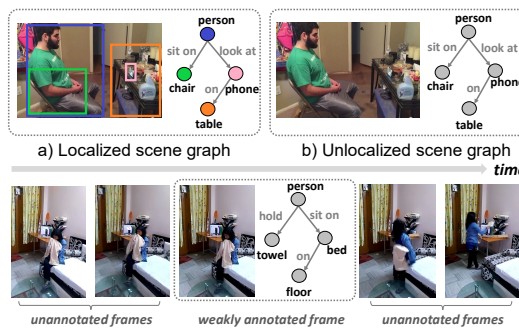

Figure 1: **(a)** Localized scene graph: It consists of all object bboxes, object categories, and predicate categories. **(b)**: Unlocalized scene graph: It consists of object and predicate categories without object bboxes. **(c)** The supervision for SF-VidSGG, which only provides an unlocalized scene graph for the middle frame.

For each video, we only need single-frame annotations instead of all-frame annotations as the fully-supervised setting (*i.e.*, reduce the number of annotated frames). 2) Frame-level: The single-frame annotation is an unlocalized scene graph (*i.e.*, avoid annotating object locations).

A straightforward solution for SF-VidSGG is: Using all the weakly annotated frames to train a weakly-supervised ImgSGG model first, and then detecting scene graphs on each frame with the ImgSGG model. Apparently, this naive ImgSGG-based method has overlooked the temporal context in the video data. To this end, we propose a novel Pseudo Label Assignment strategy **PLA**, which can serve as the first baseline for SF-VidSGG. Since PLA is agnostic to different VidSGG architectures, it can be easily incorporated into any advanced VidSGG model. Specifically, PLA decouples the problem into two steps: The first step is to assign a *pseudo* localized scene graph for every frame in the video, and the second step is to train a fully-supervised VidSGG model by the pseudo localized scene graphs. PLA consists of three modules: object pseudo label assignment module (Obj-PLA), relation pseudo label assignment module (Rel-PLA), and future predicate prediction module (FPP). In the Obj-PLA module, we detect object region proposals for all the frames. In the Rel-PLA module, we propose two relation pseudo label assignment teachers and they generate two different pseudo labels for each frame. In the FPP module, we determine adapted weights to fuse these labels from two different teachers. To effectively obtain optimal adapted weights for fusing different teacher knowledge, the FPP module exploits the temporal context based on the relation transition in videos. The relation transition means how the predicates change between the same subject-object pairs in different frames.

In summary, we make three main technique contributions in this paper:

1. We propose the first weakly-supervised VidSGG task. Compared to its fully-supervised counterpart, we try to mitigate the intensive annotations from both video-level and frame-level.
2. We propose a novel method PLA for SF-VidSGG. It utilizes two teachers to assign pseudo label for unlabeled data and refines the pseudo labels from both teachers by knowledge distillation.
3. We propose a future predicate prediction module that leverages temporal dependencies in video.

## 2  RELATED WORK

**Image Scene Graph Generation (ImgSGG).** ImgSGG aims to generate semantic graph structures — scene graphs — as the representation of images. In each scene graph, every node represents an object instance and every edge represents a visual relation between two objects. Early ImgSGG methods always directly predict all pairwise visual relations (Lu et al., 2016; Zhang et al., 2017).

Later, ImgSGG methods mainly focus on designing different context encoding architectures, such as message passing (Xu et al., 2017; Li et al., 2017), recurrent network (Zellers et al., 2018), tree-structure model (Tang et al., 2019), agent communication (Chen et al., 2019), and Transformer-based models (Lu et al., 2021). Recently, a surge of ImgSGG work starts to explore the long-tailed issue in predicate or triplet classification. Existing methods for debiased SGG can be roughly devided to four types: class-aware re-sampling (Li et al., 2021; 2022d; Desai et al., 2021), loss re-weighting (Lin et al., 2017; Yan et al., 2020; Knyazev et al., 2020), biased-model-based (Tang et al., 2020; Chiou et al., 2021; Yu et al., 2021) and noisy label correction (Li et al., 2022a;b).

**Video Scene Graph Generation (VidSGG).** Beyond static images, VidSGG aims to detect dynamic visual relations in the videos. Compared to ImgSGG, VidSGG is more challenging because they need to consider the spatio-temporal context in adjacent frames. Based on the formats of relation triplet annotations, existing VidSGG work can be categorized into two groups: 1) **Tracklet-based**: Each graph node is an object tracklet in a video clip (Shang et al., 2017; 2019) or the whole video (Liu et al., 2020; Gao et al., 2021; 2022; 2023). 2) **Frame-based**: Each graph node is an object bbox as in ImgSGG, but these visual relation triplets are dynamic in the whole video sequence (Ji et al., 2020; Feng et al., 2021; Cong et al., 2021; Li et al., 2022e). In this paper, we follow the frame-based VidSGG setting, and propose the first weakly-supervised VidSGG setting and model.

**Weakly-Supervised ImgSGG.** A ground-truth visual relation triplet annotation consists of two object locations, their object categories, and pairwise visual relations. Thus, it is extremely labor-intensive to obtain large-scale fully-annotated scene graph benchmarks (Zhang et al., 2020; Liu et al., 2022; Song et al., 2023; Liu et al., 2016). To reduce the labeling costs, several ImgSGG work proposed to generate scene graphs under weak supervision. Currently, there are two main types of weak supervision for ImgSGG: 1) **Unlocalized Scene Graphs** (Zareian et al., 2020; Shi et al., 2021; Li et al., 2022c): It only consists of image-level labels that describe the object and visual relation categories without object locations. 2) **Aligned Captions** (Shi et al., 2021; Ye & Kovashka, 2021): It provides an entailment caption for the given image. In this paper, we only utilize an unlocalized scene graph of a middle frame as the supervision for VidSGG.

**Knowledge Distillation (KD).** KD is first used for model compression (Hinton et al., 2015). Then, KD becomes a prevalent method to transfer knowledge from a (larger) teacher model to a (smaller) student model. Benefiting from the soft targets generated by teacher models, the student model can even outperform their teachers through appropriate training strategies (Zhang et al., 2018; Furlanello et al., 2018). However, these traditional KD methods are single-teacher. To reduce the limitation of data diversity and single-teacher knowledge, recent work tries to distill knowledge from multiple teachers. The idea of multi-teacher KD has been widely applied to numerous vision tasks, *i.e.*, object detection (Chen et al., 2017; Wang et al., 2019), visual question answering (Niu & Zhang, 2021; Chen et al., 2022) and ImgSGG (Li et al., 2022c). In this paper, we propose two relation pseudo label assignment teachers and use multi-teacher KD to fuse the knowledge of multi-teachers.

## 3 PROPOSED APPROACH

**Task Definition.** In this paper, we propose a new VidSGG task: *weakly-supervised VidSGG with single-frame weak supervision* (SF-VidSGG). Specifically, given an input video $V = \{I^1, ..., I^T\}$ with $T$ frames, SF-VidSGG aims to generate a scene graph $G^t$ for each frame $I^t$, and then stack them along the time axis to obtain the final video scene graph $\mathcal{G} = \{G^1, ..., G^T\}$, where $G^t = (\mathcal{N}^t, \mathcal{E}^t)$, $\mathcal{N}^t$ and $\mathcal{E}^t$ denote the set of graph nodes and edges in $G^t$, respectively. In the training stage, we only use the weakly-supervision as mentioned in Sec. 1, *i.e.*, the only supervision for SF-VidSGG is an unlocalized scene graph for a middle frame of the video $V$, denoted as $\widetilde{G}^m = (\widetilde{\mathcal{N}}^m, \widetilde{\mathcal{E}}^m)$, and $I^m$ is the only frame in the video with annotation.

### 3.1 MODEL OVERVIEW

To tackle the SF-VidSGG task, we propose a novel framework: Pseudo Label Assignment (**PLA**). The whole pipeline of PLA is illustrated in Figure 2. Specifically, it consists of three components: *object pseudo label assignment* (Obj-PLA) module, *relation pseudo label assignment* (Rel-PLA) module, and *future predicate prediction* (FPP) module. In the Obj-PLA, we detect object region proposals for all the frames. For the frame with weak annotation, we further complete its unlocal-

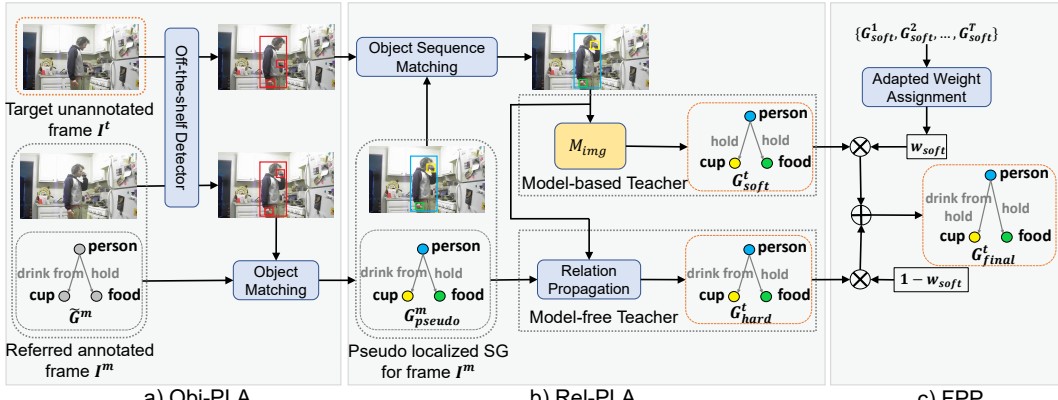

| a) Obj-PLA | b) Rel-PLA | c) FPP |

Figure 2: A overview of PLA. Here we show how to assign pseudo label for one unannotated frame $I^t$ as an example. PLA consists of three module: (a) **Obj-PLA** detects pseudo object bounding boxes and assigns pseudo category labels to them. (b) **Rel-PLA** proposes two teachers to assign pseudo relation annotations ($G^t_{\text{soft}}$ and $G^t_{\text{hard}}$). (c) **FPP** first calculates a adapted weight by $G^t_{\text{soft}}$ of all frames in the video, then calculates the final pseudo relation annotations ($G^t_{\text{final}}$) by the weighted sum. Finally, we train a fully supervised VidSGG model.

ized scene graph $\widetilde{G}$ to obtain a pseudo localized scene graph $G_{\text{pseudo}}$. In the Rel-PLA, we propose two relation pseudo label assignment teachers: model-based and model-free. The model-based teacher assigns "soft" predicate pseudo labels $\mathcal{P}_{\text{soft}}$ by an ImgSGG model, while the model-free teacher assigns "hard" predicate pseudo labels $\mathcal{P}_{\text{hard}}$ by some heuristic rules. After $\mathcal{P}_{\text{soft}}$ and $\mathcal{P}_{\text{hard}}$ are assigned, the FPP module determines adapted weights to fuse $\mathcal{P}_{\text{soft}}$ and $\mathcal{P}_{\text{hard}}$ to obtain the final pseudo label $\mathcal{P}_{\text{final}}$. Finally, we train a fully-supervised VidSGG model by $\mathcal{P}_{\text{final}}$.

## 3.2 OBJECT PESUDO LABEL ASSIGNMENT (OBJ-PLA)

In this module, we aim to generate pseudo object category annotations for all frames. For each frame $I^t$ in input video, we first use an off-the-shelf detector to generate a set of proposal $\mathcal{O}^t$. Each proposal $o^t_i \in \mathcal{O}^t$ has a corresponding bounding box position $\hat{b}^t_i$ and an initial object category $\hat{c}^t_i$, i.e., $o^t_i = (b^t_i, \hat{c}^t_i)$. Since the differences in object category ontology between the pretrained detector and VidSGG benchmarks, we then map the detected object categories to the object categories in the VidSGG dataset if there is overlapping between their synonyms in WordNet (Miller, 1995), e.g., "woman" is mapped to "person", "pizza box" is mapped to "box".

For the annotated frame $I^m$, Obj-PLA has an additional step: matching these detected bounding boxes with entities in the unlocalized scene graph to obtain a pseudo localized scene graph. Here we use a category-based strategy: a bounding box and a entity will be matched if they have the same category. This strategy has a weakness that it cannot distinguish bounding boxes with the same object category. When the detector finds multiple bounding boxes with the same object category, we just randomly choose one of them to match. To solve this problem, Li et al. (2022c) utilizes pretrained VL models to calculate relevant scores between entities with bounding boxes. The final matches are decided by the object category labels and the relevant scores. We leave these advanced methods for the future work.

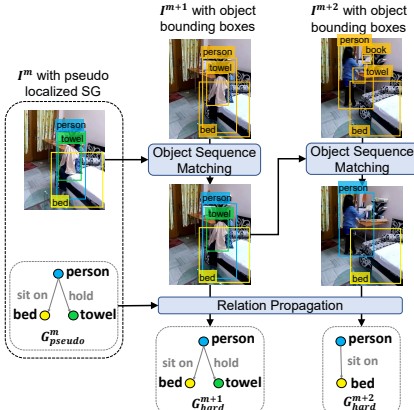

Figure 3: Illustration of model-free teacher.

## 3.3 RELATION PESUDO LABEL ASSIGNMENT (REL-PLA)

After Obj-PLA, we obtain a pseudo localize scene graph for the annotated Frame and object bboxes for all other unannotated frames. In Rel-PLA, we design two strategies (teachers) to generate pseudo predicate labels for the unannotated frames.

**Model-based Teacher.** Model-based Teacher assigns "soft" predicate pseudo labels by an ImgSGG model. It first obtains pseudo localized scene graphs $G_{\text{pseudo}}$ from the annotated frame set by Obj-PLA, then train a ImgSGG model by these scene graphs. After training, each un-annotated frame $I^t$ is fed to the network to obtain $G_{\text{soft}}^t = M_{\text{img}}(I^t)$, the scene graph for $I^t$. Here we represent the predicate in $G_{\text{soft}}^t$ by the category distribution of predicate, besides one-hot labels. Inspired by knowledge distillation, we believe that the soft pseudo label can preserve more information than the hard pseudo label (Hinton et al., 2015).

**Model-free Teacher.** Model-free teacher assigns "hard" pseudo prediacte labels by some heuristic rules. The first step is **object sequence matching**. We adopt the object category and the IoU to match the objects detected in the different frames. Two object $o_i^t$ and $o_j^{t+1}$ in adjacent frames can be matched if they have same object categories and $IoU(b_i^t, b_j^{t+1}) > \eta$, where $\eta$ is the IoU matching threshold to filter objects. We *frame-by-framely* match objects from the middle frame to both ends. As shown in Figure 3, "book" and "towel" in $I^{m+2}$ are not matched with any object in $I^{m+1}$ because there is no object detected in $I^{m+1}$ with category "book" and IoU of bounding boxes with category "towel" in $I^{m+1}$ and $I^{m+2}$ is lower than $\eta$. The second step is **relation propagation**. We simply propagate the predicate of a subject-object pair in the annotated frame to the same subject-object pair in the un-annotated frames.

### 3.4 FUTURE PREDICATE PREDICTION (FPP) MODULE

**Multi-Teacher Fusion Strategy.** After obtaining $\mathcal{P}_{\text{soft}}$ and $\mathcal{P}_{\text{hard}}$ by two teachers, the simplest fusing strategy is treating each teacher equally and averaging the two pseudo labels. However, this strategy overlooks the noise in the model-based teacher. The model-based teacher sometimes assign wrong pseudo labels due to the incorrect predictions in model $M_{\text{img}}$. On the other hand, pseudo labels assigned by the model-free teacher can be seen as the default predictions because these labels have already appeared in the video. To some extent, the model-free teacher can be seen as a complement when the model-based teacher assign a wrong pseudo label. Therefore, a better strategy is assigning adaptive weights for $\mathcal{P}_{\text{soft}}$ and $\mathcal{P}_{\text{hard}}$ and then calculating the final pseudo labels by the weighted sum:

$$\mathcal{P}_{\text{final}}^i = w_{\text{soft}}^i * \mathcal{P}_{\text{soft}}^i + w_{\text{hard}}^i * \mathcal{P}_{\text{hard}}^i, \tag{1}$$

where $w_{\text{soft}}^i$ and $w_{\text{hard}}^i$ denotes the weights of the soft and hard predicate labels for video $i$, respectively. In the FPP module, we calculate $w_{\text{soft}}^i$ and $w_{\text{hard}}^i$ by regularity of relation transition.

**Relation Transition.** Relation transition means how the predicates change between the same subject-object pairs in different frames. Mi et al. (2021) has proved that relation transition has the temporal tendency, which means we can infer their relation in the next frame according to their relation in the current frame for the same subject-object pair. As the example shown in Figure 4, we can find two main conclusions: 1) Relation transitions have some regular patterns, *e.g.*, for subject-object pair "people-chair", relation "leaning on" is mainly transferred to "sitting on", but hardly transferred to "lying on" and "standing on". 2) Relation transitions with different subject-object pairs have different regular patterns, *e.g.*, the statistical distributions for "people-chair" and "people-food" are different. Therefore, we can evaluate the quality of predicate pseudo labels by regularity of relation transition in video. Specifically, irregular relation transitions (*e.g.*, "leaning on" $\rightarrow$ "lying on" for "people-chair") are of low quality, then we assign low weights to them.

**Proposed FPP.** As shown in Figure 5, We take two steps to calculate weights $w_{\text{soft}}$ and $w_{\text{hard}}$. 1) **Predicate prediction**: For each predicate $p^t$ in $I^t$ assigned by the model-based teacher, we predict $\hat{p}^{t+1}$ as the predicate that $p^t$ will transition to in $I^{t+1}$. For each $\hat{p}^{t+1}$, we find its corresponding predicate $p^{t+1}$ in $I^{t+1}$ that actually assigned by the model-based teacher. 2) **Adapted weight assignment**: We first calculate KL-divergence between $\hat{p}^{t+1}$ and $p^{t+1}$ as the inconsistency score. Then, we assign low weights to pseudo labels with high inconsistency scores and high weights to pseudo labels with low inconsistency scores. In the following, we detailed introduce these two steps.

*Predicate Prediction.* To predict the future predicates, we construct a relation transition matrix $T \in \mathbb{R}^{N_o * N_r * N_r}$, where $N_o$, $N_r$ are the number of objects and relations in the dataset, respectively. The relation transition matrix indicates the statistical distribution of relation transition in the VidSGG dataset. Given a predicate $p^t$ in frame $I^t$ with object category $c_o$, we predict $\hat{p}^{t+1}$ as:

$$\hat{p}^{t+1} = p^t \times T[id(c_o)], \tag{2}$$

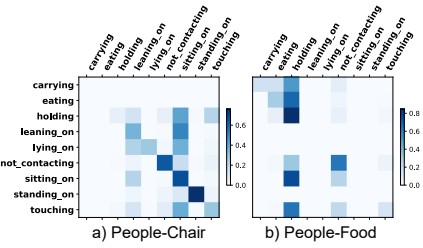

Figure 4: Statistical distributions of relation transition in AG. Each row represents the distribution of relations in the next frame for the relation in the left.

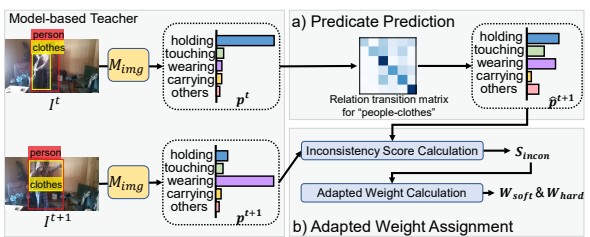

Figure 5: Illustration of the FPP module, which consists of two parts: a) predicate prediction, b) adapted weight assignment.

where $id(c_o)$ is the id of $c_o$ in the vocabulary $C_o^g$. Then, for each $\hat{p}^{t+1}$, we try to find its corresponding predicate $p^{t+1}$ in $I^{t+1}$ that actually assigned by the model-based teacher. Sometimes $p^{t+1}$ is not existed because the detector can not detect the subject-object pair of $\hat{p}^{t+1}$ in frame $I^{t+1}$. To handle this problem, we adopt the object category and the IoU to filter predicates without corresponding subject-object pair in the next frame, as we did in object sequence matching of the model-free teacher. The set of $\hat{p}^{t+1}$ and $p^{t+1}$ after filtering in the whole video are denoted as $\hat{\mathcal{P}}_{\text{filter}}$ and $\mathcal{P}_{\text{filter}}$, respectively.

***Adapted Weight Assignment.*** We first calculate the inconsistency score by KL-divergence as below:

$$S_{\text{incon}} = \text{KL}(\hat{\mathcal{P}}_{\text{filter}} || \mathcal{P}_{\text{filter}}). \tag{3}$$

Then, we calculate the inconsistency score for $\mathcal{P}_{\text{soft}}$, denoted as $S_{\text{incon}}^{\text{soft}}$. A high inconsistency score indicates that these predicate pseudo labels do not satisfy the regularity of relation transition, so we should assign low weights to them. To the end, we define the function as:

$$w_{\text{soft}} = 2 - 2 * \text{Sigmoid}(S_{\text{incon}}^{\text{soft}}), \quad w_{\text{hard}} = 1 - w_{\text{soft}}. \tag{4}$$

## 3.5 Training Objectives

**FPP module.** We assume that the relation transition matrix $T$ in FPP module indicates the statistical distribution of relation transition in the VidSGG dataset. However, in the SF-VidSGG task, we can not count the distribution of relation transition directly because we only have single-frame weak supervision for each video. An alternative approach is generating pseudo labels by the model-based teacher and then learning the distribution of these pseudo labels. To accomplish this, we formulate the loss function as below to train the FPP module:

$$\mathcal{L}_{\text{FPP}} = \text{KL}(\hat{\mathcal{P}}_{\text{filter}}^{\text{soft}} || \mathcal{P}_{\text{filter}}^{\text{soft}}), \tag{5}$$

**Student Model.** We adopt the KL-divergence loss for training like most knowledge distillation methods, which can be formulated as follows:

$$\mathcal{L}_{\text{KD}} = \text{KL}(\mathcal{P}_{\text{pred}} || \mathcal{P}_{\text{final}}), \tag{6}$$

where $\mathcal{P}_{pred}$ denotes the output of the student model, and $\mathcal{P}_{final}$ denotes the predicate pseudo labels.

Finally, the total training loss of PLA is:

$$\mathcal{L} = \mathcal{L}_{\text{KD}} + \mathcal{L}_{\text{FPP}}. \tag{7}$$

## 4 Experiments

Due to the limited space, the details about experimental settings (including the evaluation dataset and metrics) and implementation details are left in the appendix.

### 4.1 Ablation Studies

In this section, we validated the effectiveness of our PLA by answering the following questions: **Q1**: How does each component in PLA contribute to the performance on the SF-VidSGG task? **Q2**:

Table 1: Performance (%) on dataset Action Genome of models with different components of PLA. Model `A` is the baseline, `B` only uses model-based teacher, `C` only uses model-free teacher, `D` uses both teachers and treats them equally, `E` uses both teachers and set adapted weights by FPP module.

| Task | # | Model | Components | | | | With Constraint | | | No Constraint | | |
|------|---|-------|---------|-------------|-----------|-----|------|------|------|------|------|------|
| | | | Obj-PLA | Model-based | Model-free | FPP | R@10 | R@20 | R@50 | R@10 | R@20 | R@50 |
| SGDet | A | STTran† | ✓ | ✗ | ✗ | ✗ | 14.32 | 20.42 | 25.43 | 14.78 | 21.72 | 30.87 |
| | B | STTran | ✓ | ✓ | ✗ | ✗ | 14.99 | 21.11 | 26.12 | 15.46 | 22.56 | **31.75** |
| | C | STTran | ✓ | ✗ | ✓ | ✗ | 14.69 | 20.71 | 25.70 | 15.09 | 22.04 | 31.26 |
| | D | STTran | ✓ | ✓ | ✓ | ✗ | 15.11 | 21.16 | 26.03 | 15.57 | 22.56 | 31.60 |
| | E | STTran | ✓ | ✓ | ✓ | ✓ | **15.39** | **21.44** | **26.24** | **15.83** | **22.83** | 31.74 |
| SGCls | A | STTran† | ✓ | ✗ | ✗ | ✗ | 35.19 | 35.80 | 35.80 | 44.25 | 51.76 | 55.80 |
| | B | STTran | ✓ | ✓ | ✗ | ✗ | 36.58 | 37.19 | 37.20 | 45.54 | 52.52 | **56.46** |
| | C | STTran | ✓ | ✗ | ✓ | ✗ | 35.70 | 36.32 | 36.32 | 45.44 | 52.82 | 56.07 |
| | D | STTran | ✓ | ✓ | ✓ | ✗ | 36.48 | 37.06 | 37.06 | 44.52 | 52.52 | 56.17 |
| | E | STTran | ✓ | ✓ | ✓ | ✓ | **36.81** | **37.46** | **37.46** | **46.00** | **53.01** | 56.42 |
| PredCls | A | STTran† | ✓ | ✗ | ✗ | ✗ | 59.58 | 61.13 | 61.13 | 74.86 | 89.91 | 98.26 |
| | B | STTran | ✓ | ✓ | ✗ | ✗ | 61.23 | 62.79 | 62.79 | 76.08 | 89.91 | 97.92 |
| | C | STTran | ✓ | ✗ | ✓ | ✗ | 61.26 | 62.93 | 62.93 | 77.09 | **91.99** | **99.00** |
| | D | STTran | ✓ | ✓ | ✓ | ✗ | 61.19 | 62.69 | 62.70 | 76.51 | 90.82 | 98.46 |
| | E | STTran | ✓ | ✓ | ✓ | ✓ | **61.64** | **63.25** | **63.25** | **77.20** | 91.53 | 98.61 |

Table 2: Comparison (%) with other state-of-the-art fully-supervised approach.

| Supervision | Model | PLA | With Constraint | | | No Constraint | | |
|-------------|-------|-----|------|------|------|------|------|------|
| | | | R@10 | R@20 | R@50 | R@10 | R@20 | R@50 |
| Weak | STTran Cong et al. (2021) | ✓ | 15.4 | **21.4** | **26.2** | 15.8 | **22.8** | 31.7 |
| | DSG-DETR Feng et al. (2021) | ✓ | **15.5** | 21.3 | 25.9 | **15.9** | 22.7 | **31.9** |
| Full | STTran Cong et al. (2021) | ✗ | 25.2 | 34.1 | 37.0 | 24.6 | 36.2 | 48.8 |
| | APT Li et al. (2022e) | ✗ | 26.3 | **36.1** | **38.3** | 25.7 | 37.9 | **50.1** |
| | DSG-DETR Feng et al. (2021) | ✗ | **30.3** | 34.8 | 36.1 | **32.1** | **40.9** | 48.3 |

How far are the gaps between PLA and fully-supervised VidSGG approaches? **Q3**: Can a stronger student model help to improve the performance of PLA? **Q4**: What is the upper-bound under our pseudo label assignment training paradigm? **Q5**: What is the influence of different annotation frame choosing strategies?

### 4.1.1 EFFECTIVENESS OF EACH COMPONENT (Q1)

**Settings.** To analyze the importance of each component in PLA, we implemented the straightforward model mentioned in Sec. 1 as the baseline model. It first obtained pseudo localized scene graphs $G_{\text{pseudo}}$ from the annotated frame set by Obj-PLA, then trained a STTran† model by these scene graphs. The gray lines in Table 1 show the results of the baseline model (model `A`). We also trained the models with different components of PLA, which is shown in Table 1.

**Effectiveness of Rel-PLA.** Compared to the baseline model `A`, models with model-based teacher or model-free teacher (model `B` and model `C`) has a significant improvement. The experimental results demonstrate that by utilizing the unannotated frames in the video, model `B` and model `C` are able to outperform the baseline model `A` across almost all the metrics, only except for no constrained PredCls-R@50 of model `B`. Particularly, model `B` and model `C` have outperformed the baseline by 5.24% (15.07 *v.s.* 14.32) and 2.58% (14.69 *v.s.* 14.32) relatively in terms of R@10 with constraint criteria for SGDet, respectively. This is owing to that the two teachers have the ability to assign high-quality pseudo labels to the unannotated frames.

**Effectiveness of FPP.** As shown in Table 1, model `D` and model `E` are all use both teachers, but model `E` uses the FPP module to set adapted weights. We can observe that model `E` outperforms

model D across all the metrics. Especially, model E has achieved about 2% relative improvement over model D on R@10 metric under the no constraint criteria for SGDet. The improvement proves that FFP has the ability to assign more accurate pseudo labels than the simple fixed weight strategy.

Table 3: Performance (%) for SGDet on AG of baseline (with obj-PLA), GTRel, and PLA.

| Method | With Constraint | | | No Constraint | | |
|---|---|---|---|---|---|---|
| | R@10 | R@20 | R@50 | R@10 | R@20 | R@50 |
| Baseline | 14.32 | 20.42 | 25.43 | 14.78 | 21.72 | 30.87 |
| PLA | 15.39 | 21.44 | 26.24 | 15.83 | 22.83 | 31.74 |
| GTRel | **16.10** | **22.38** | **27.33** | **16.73** | **24.02** | **33.40** |

Table 4: Ablation studies (%) on different annotation frame-choosing strategies.

| Strategy | With Constraint | | | No Constraint | | |
|---|---|---|---|---|---|---|
| | R@10 | R@20 | R@50 | R@10 | R@20 | R@50 |
| First | 14.38 | 20.45 | 25.60 | 14.85 | 21.73 | 30.68 |
| Last | 14.25 | 20.46 | 25.66 | 14.67 | 21.67 | 30.72 |
| Random | 14.91 | 20.87 | 25.82 | 15.48 | 22.40 | 31.59 |
| Middle | **15.39** | **21.44** | **26.24** | **15.83** | **22.83** | **31.74** |

### 4.1.2 GAP BETWEEN PLA AND FULLY SUPERVISED APPROACHS (Q2)

**Settings.** We compared our PLA with three state-of-the-art fully-supervised VidSGG methods: STTran (Cong et al., 2021), APT (Li et al., 2022e), and DSG-DETR (Feng et al., 2021).

**Results.** As shown in Table 2, we can observe that there is still a gap between the performance of PLA and other fully-supervised approaches (*e.g.*, 15.4 *v.s.*25.2 for STTran in terms of R@10 with constraint criteria for SGDet). However, considering that the annotation cost of our method is extremely lower ($<5\%$) than that of full supervision. we think it is still a good start for weakly supervised frame-level VidSGG. We leave the task to reduce the gap as the future work.

### 4.1.3 EFFECTS OF DIFFERENT STUDENT MODELS (Q3)

**Settings.** We used DSG-DETR as the student model of PLA. Since PLA is agnostic to the specific VidSGG architecture, it can be easily incorporated into any advanced VidSGG models. We used the same teachers as for STTran to generate pseudo labels for DSG-DETR, then trained a fully-supervised VidSGG model by these pseudo labels.

**Results.** As shown in Table 2, PLA with DSG-DETR also achieves similar performance with STTran across all the metrics. These results show that PLA is a general framework and can easily extend to advanced fully supervised VidSGG methods.

### 4.1.4 BEST CASE: GROUND-TRUTH UNLOCALIZED SCENE GRAPHS (Q4)

**Settings.** Like most existing SGG methods, PLA has two main steps: the first step is to detect objects in each frame, and the second step is to predict relations between these objects. The performance of PLA is mainly affected by the first step due to the poor performance of the off-shelf detector in Action Genome. Under this object detection strategy, the upper-bound performance is not high. We can train an upper-bound model as follows: first detect objects by Obj-PLA, then assign ground truth relation labels between them, finally use these labels to train the student model. We named this upper-bound model as GTRel.

**Results.** As shown in Table 3, we observe that GTRel significantly improves the performance over the baseline. Specifically, GTRel has outperformed the baseline by 13.19% (16.73 *v.s.*14.72) relatively in terms of R@10 no constraint criteria for SGDet. We also demonstrate that PLA has ability to reduce the gap between the baseline model and GTRel. For example, GTRel improves by 12.43% (16.10 *v.s.*14.32) relatively over baseline in terms of R@10 with constraint criteria for SGDet, but only improves by 4.61% (16.10 *v.s.*15.39) relatively over PLA on same metric. These results show that PLA is able to reduce about half performance gap between the baseline model and the upper-bound model GTRel for the SF-VidSGG task. Besides, the method to detect objects under unlocalized scene graph setting is not our main contribution. As a general framework, if better methods to detect objects are proposed, PLA can easily extend to them.

### 4.1.5 EFFECTS OF DIFFERENT ANNOTATION FRAME-CHOOSING STRATEGIES. (Q5)

**Settings.** We compared four simple annotation frame choosing strategies. Specifically, "First", "Last", and "Middle" means annotating the "first", "last", and "middle" frame with an unlocalized scene graph in each video, respectively. And "Random" means choosing one frame randomly.

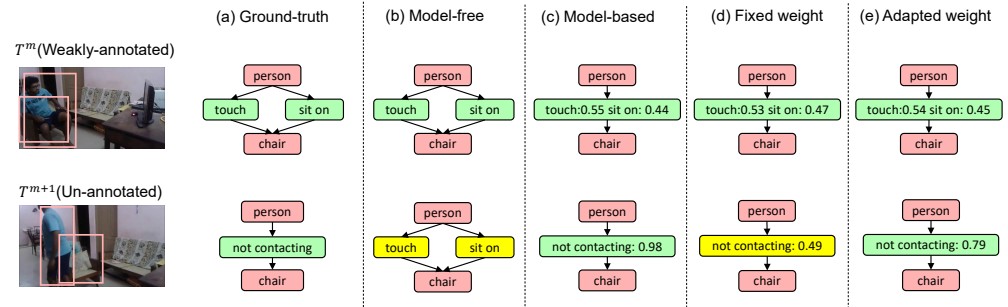

Figure 6: Visualization results of pseudo labels assigned by: (a) ground-truth, (b) model-based teacher, (c) model-free teacher, (d) both teachers with fixed weights, (e) both teachers with adapted weights. (a) and (b) are hard labels (each node indicates a single relation), while (c) - (e) are soft labels (each node indicates a relation distribution). The green relations mean correct predictions, the yellow relations are wrong predictions that do not exist in the ground-truth.

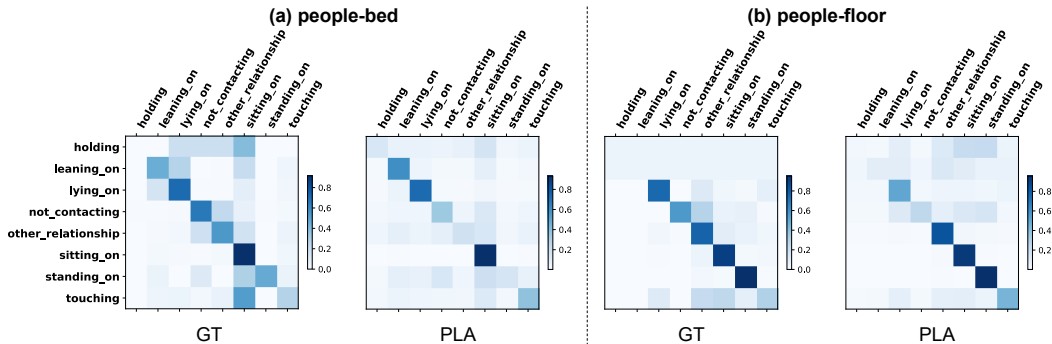

Figure 7: Qualitative results for the statistical distributions of relation transition. In each subfigure, the left is the ground-truth result and the right is result of PLA.

**Results.** As shown in Table 4, We can observe that model trained by the middle annotation frame has the best performance, eg., in terms of R@10 with constraint criteria for SGDet, strategy "Middle" outperforms "First", "Last", and "Random" by 1.01, 1.14, and 0.48, respectively. It also proves that the middle frame usually contains more information.

## 4.2 QUALITATIVE RESULTS

**Pseudo Label Visualization.** Figure 6 shows a qualitative comparison of the pseudo labels generated by different variants of PLA. Results indicates that the adapted weights given by FPP module can generate more accurate pseudo labels than fixed weights. Specifically, relation in $T^{m+1}$ ("not contacting") is different from that in $T^m$ ("touching" and "sitting on"), so the model-free teacher give totally wrong relations ("touching" and "sitting on") and the model-based teacher predict an almost accurate relation (98% confidence for "not contacting"). Compared to fixed weight strategy, the FPP module gives more weight to the model-based teacher (79% $v.s.$49% confidence for "not contacting") and generates more accurate pseudo labels.

**Relation Transition Distribution Visualization.** Figure 7 demonstrates the statistical distributions of relation transition counted by ground truth and learned by PLA. From Figure 7, we can observe that PLA can learn a distribution similar to the ground truth.

## 5 CONCLUSIONS

In this paper, we presented the first weakly-supervised VidSGG task with only single-frame weak supervision: SF-VidSGG. Unlike VidSGG, SF-VidSGG only require a weak annotaion without bounding box for one frame in each video. To the end, we proposed a novel and efficient method named PLA for SF-VidSGG, which based on pseudo label assignment for the video. We validated the effectiveness of each component of PLA through extensive experiments. In future work, we would like to explore weaker supervision in VidSGG or more accurate pseudo label assignment.

**Ethics Statement**

Video scene generation models may be utilized in unauthorized human monitoring and surveillance, which leads to some ethical and privacy issues. Apart from these general issues that already exist in the video scene graph generation task, our paper has no additional ethical issues.

**Reproducibility Statement**

PLA is mainly implemented based on the released code of STTran (Cong et al., 2021) and evaluated on the Action Genome (AG) (Ji et al., 2020) dataset, which is publicly available with a license that allows free usage for research purposes. We will also release our code of PLA.

**Acknowledgment**

This work was supported by the National Key Research & Development Project of China (2021ZD0110700), the National Natural Science Foundation of China (U19B2043, 61976185), and the Fundamental Research Funds for the Central Universities (226-2022-00051).

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

# A    APPENDIX

This Appendix has the following contents:

- The details about the valuation dataset are given in Sec. A.1
- The evaluation metrics are given in Sec. A.2.
- The implementation details are given in Sec. A.3.
- The detailed hyperparameter setting are given in Sec. A.4.
- Ablation studies on imperfect object detector in Sec. A.5.
- Ablation studies on the different proportions of the weakly supervised labels in Sec. A.6.
- More qualitative results are in Sec. A.7.

## A.1    DATASET

We evaluated our method PLA on the challenging VidSGG benchmark: Action Genome (AG) (Ji et al., 2020). AG is a prevalent VidSGG dataset providing human-object relationships with frame-level annotations. Specifically, AG consists of 9,201 videos, which annotated for 234,253 frames by 476,299 bounding boxes of 36 object categories and 1,715,568 instances of 25 relation categories. All relation categories in AG are split into three types: 3 attention relations, 6 spatial relations and 16 contact relations. Each subject-object pair has one attention relation but may has multiple spatial or contact relations. In our SF-VidSGG setting, we used the official splits as fully-supervised work, *i.e.*, 7,464 videos for training, 1,737 videos for testing. For each video in the training set, we only used the unlocalized scene graph of the middle frame as supervision.

## A.2    EVALUATION METRICS

Following the fully-supervised setting, we evaluated PLA on three different settings: 1) *Scene Graph Detection* (**SGDet**): Given a video, the model needs to detect the objects and predict predicate categories of object pairs for all frame. In SGDet, an object is considered to be successfully detected if the IoU between the predicted bbox and its ground-truth bbox is larger than 0.5. 2) *Scene Graph Classification* (**SGCls**): Given a video and object bboxes, the model needs to predict object and predicate categories for all frames. 3) *Predicate Classification* (**PredCls**): Given a video, object bboxes and object labels, the model needs to predict predicate categories for all frames. Since the ground-truth bboxes are not provided in SF-VidSGG, we only trained models under **SGDet** settings, then evaluated them under these three settings. We used Recall@K (**R@K**, K = [10,20,50]) as our evaluation metric, which measures the ratio of the ground-truth relation triplets among the top-K predicted relation triplets. Furthermore, we adopt two typical strategies to generate dynamic scene graphs: 1) **With Constraint**: It only allows each subject-object pair to have at most one predicate. 2) **No Constraint**: It allows each subject-object pair to have multiple predicates.

## A.3 IMPLEMENTATION DETAILS

In the Obj-PLA module, we used the pretrained VinVL (Li et al., 2020; Zhang et al., 2021) with backbone ResNeXt-152 C4 as the off-the-shelf detector. Specifically, it is pre-trained on multiple detection benchmarks, including COCO (Lin et al., 2014), OpenImages (Kuznetsova et al., 2020), Objects365 (Shao et al., 2019), and Visual Genome (Krishna et al., 2017). This detector is capable of detecting 1,594 general object categories. We kept objects which confidence is higher than 0.2 and extracted the 2048-D region features from the detector. In the Rel-PLA module, we modified STTran (Cong et al., 2021) to an ImgSGG model that process one frame at a time, denoted as STTran$^\dagger$. We used STTran$^\dagger$ as the teacher model to assign pseudo soft labels in Rel-PLA. For the model-free teacher, we set the IOU matching threshold $\eta = 0.5$. In FPP module, we set the initial learning rate $1e^{-3}$. We used STTran as the student model, and followed the same training settings (*e.g.*, learning rate and batch size) of the original paper.

## A.4 HYPERPARAMETER SETTINGS

We analyzed the influence of the hyperparameter $\eta$, which denote the IoU matching threshold in Rel-PLA. We set different $\eta$ for the model only uses model-free teacher (model C). All results are in Table 5. Particularly, when the threshold $\eta$ is set to 0, we match the objects in different frames only by the object category without the IoU of the bounding boxes in adjacent frames.

Table 5: Different size of the weakly supervised labels.

| $\eta$ | With Constraint | | | No Constraint | | |
|---|---|---|---|---|---|---|
| | R@10 | R@20 | R@50 | R@10 | R@20 | R@50 |
| 0.0 | 13.58 | 19.52 | 24.62 | 14.02 | 20.95 | 30.49 |
| 0.2 | 14.64 | 20.54 | 25.33 | **15.11** | 22.03 | 31.11 |
| 0.5 | **14.69** | **20.71** | **25.70** | 15.09 | **22.04** | **31.26** |

**Results.** As shown in Table 5, we can observe that the model with $\eta = 0$ has a weak performance due to more wrong objects matching without IoU information. And the model with $\eta = 0.5$ slightly outperforms the model with $\eta = 0.2$. Therefore, we set $\eta = 0.5$ for all experiments.

Table 6: Ablation studies on imperfect object detector.

| Dropped | With Constraint | | | No Constraint | | |
|---|---|---|---|---|---|---|
| Bboxes | R@10 | R@20 | R@50 | R@10 | R@20 | R@50 |
| 0% | **15.39** | **21.44** | **26.24** | **15.83** | **22.83** | **31.74** |
| 10% | 14.16 | 19.86 | 23.60 | 14.66 | 21.12 | 29.48 |
| 20% | 13.18 | 17.85 | 20.99 | 13.66 | 19.46 | 27.26 |
| 40% | 11.48 | 14.86 | 16.53 | 12.01 | 22.14 | 24.72 |
| 80% | 4.84 | 5.29 | 5.32 | 5.62 | 7.15 | 8.48 |

Table 7: Ablation studies on different proportion of the weakly supervised labels.

| Proportion | With Constraint | | | No Constraint | | |
|---|---|---|---|---|---|---|
| | R@10 | R@20 | R@50 | R@10 | R@20 | R@50 |
| 1 frame | 15.39 | 21.44 | 26.24 | 15.83 | 22.83 | 31.74 |
| 10% | 15.10 | 21.17 | 25.96 | 15.55 | 22.56 | 31.88 |
| 20% | 16.00 | 21.91 | 26.61 | 16.56 | 23.55 | 32.89 |
| 40% | 16.23 | 22.50 | 27.37 | 16.83 | 24.07 | **33.30** |
| 80% | **16.26** | **22.57** | **27.68** | **16.89** | **24.18** | 33.26 |

## A.5 EFFECTS OF IMPERFECT OBJECT DETECTORS

**Settings.** To simulate an imperfect object detector, we randomly dropped a certain proportion of bounding boxes given by the object detector. Without loss of generality, we set five dropped proportion as 0%, 10%, 20%, 40% and 80%.

**Results.** As shown in Table 6, we can observe that the bigger dropped proportion, the worse the performance of PLA, eg. the setting with 0% dropped proportion outperforms settings with 10%, 20%, 40% and 80% dropped proportion by 1.23, 2.21, 3.91 and 10.55, respectively. When dropping 10%, 20% and 40% bboxes, the performance has not decreased much, but the performance decreased significantly if dropping 80% bboxes. Results show that PLA is affected by the preformance of the object detector, but it is the weakness for all two-stage (first detect objects, then predict relations between them) SGG method.

### A.6 EFFECTS OF THE SIZE OF WEAKLY SUPERVISED LABELS

**Settings.** Although PLA is designed for single-frame weak supervision, it can be easily extended to multi-frame weak supervision. For the model-based teacher, we just train the model with more frames. For the model-free teacher, we take each annotated frame as a starting point to propagate relations. Without loss of generality, we set four annotated proportion as 10%, 20%, 40% and 80%. We annotated frame evenly in temporal axis, for example, if a 20-frame video with 20% annotated proportion, we will annotate its 1st, 7th, 14th and 20th frame.

**Results.** As shown in 7, we can observe that the more annotated frames, the better the performance of PLA, e.g., model with 80% annotation frames outperforms models with 10%, 20% and 40% annotation frames 1.16, 0.26 and 0.03. It proves that PLA can easily adapt to multi-frame weak supervision setting.

### A.7 MORE QUALITATIVE RESULTS ON ACTION GENOME DATASET

More qualitative results of PLA on Action Genome are shown in Figure 8. From Figure 8, we can observe that PLA has the ability to detect dynamic relations, *e.g.*, in the third video, the relation between person and table from "not contacting" to "touching".

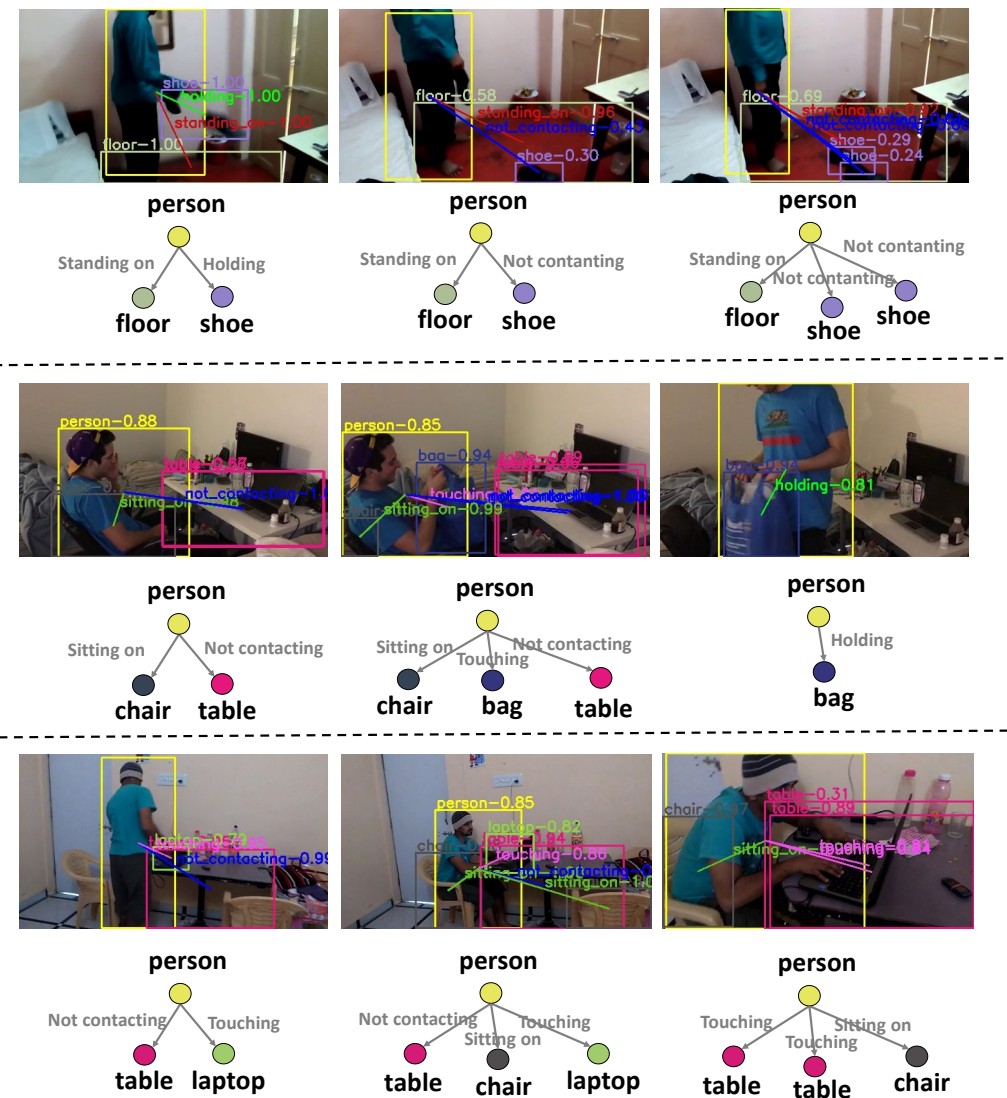

Figure 8: More qualitative results on Action Genome dataset.

