# OpenReview forum: "Video Scene Graph Generation from Single-Frame Weak Supervision"
_ICLR.cc/2023/Conference — ICLR 2023 poster_

### Official Review · Reviewer_Ca8D · 2022-10-23

**Confidence:** 4
**Correctness:** 3
**Technical Novelty And Significance:** 3
**Empirical Novelty And Significance:** 3
**Recommendation:** 8

**Clarity, Quality, Novelty And Reproducibility:**

This paper is well-written and is easy to follow
This paper proposes a new task for VidSGG. It is meaningful for real-world application. The proposed method seems work and interesting.

**Strength And Weaknesses:**

Strengths
1)The proposed SF-VidSGG task is interesting and meaningful. The weakly supervised scene that this task focuses on are more in line with real-world cases and thus help bridge the gap between video scene graph generation research and practical applications.
2)The proposed PLA method is novel and model-agnostic. In particular, it generates different “pseudo labels” using multiple-teachers, and dynamically assigns weights for these teachers by considering the temporal dependencies in videos.

Weaknesses
The paper does not give an exact reason for choosing the “single-frame weak supervision”. It is required to provide a comprehensive discussion on different choices of weak supervision and explain why not using other types of weak supervision.

**Summary Of The Paper:**

This paper deals with the video scene graph generation problem. 1) this paper proposes a new task named SF-VidSGG, which requires to train the VidSGG models with only weak supervision. 2) this paper proposes a novel pseudo-label assignment (PLA) method for the SF-VidSGG task. In particular, PLA decouples the problem into two steps: 1) assigning a pseudo-localized scene graph to each frame in the video, and 2) training a VidSGG model in a fully supervised manner using the pseudo-localized scene graph. Extensive experiments show the effectiveness.

**Summary Of The Review:**

This paper proposes a weakly-supervised VidSGG task. It utilizes the unlocalized scene graph of intermediate frames and proposes a model-agnostic pseudo label assignment method PLA. The experiments showthe effectiveness.

---

> ### Author Response · Authors · 2022-11-18
> **Response to Reviewer Ca8D**
>
> Thank you for the detailed comments. We are willing to address all your questions.
>
> ## Q1: Why we choose the “single-frame weak supervision”?
>
> > It is required to provide a comprehensive discussion on different choices of weak supervision and explain why not using other types of weak supervision.
>
> **A1**: We choose the “single-frame weak supervision” from two aspects.
>
> - We aim to minimize labeling costs in the VidSGG task, so the SF-VidSGG task tries to relieve the intensive annotation issues from two levels: 1) Video-level: For each video, we only need single-frame annotations instead of all-frame annotations as the fully-supervised setting (i.e., reduce the number of annotated frames). 2) Frame-level: The single frame annotation is an unlocalized scene graph (i.e., avoid annotating object locations).
>
> - Another setting is what annotation frame-choosing strategy should we select. In our paper, we choose the middle frame as the annotation frame. Since the main event of the video is usually in the middle, we think the model trained with middle-annotated frames has the best performance.
>
> Following your comments, we added experiments of different annotation frame choosing strategies for the sake of thoroughness.
>
> **Settings**: We designed four simple annotation frame choosing strategies. "First" means annotating the first frame with an unlocalized scene graph in each video, "Last" means choosing the last frame, "Random" means choosing one frame randomly, and "Middle" means choosing the middle frame.
>
> **Results**: As shown in Table below, We can observe that model trained by the middle annotation frame has the best performance, eg., in terms of R@10 with constraint criteria for SGDet, strategy "Middle" outperforms "First", "Last" and "Random" by 1.01%, 1.14% and 0.48%, respectively. So the conclusion is annotating the middle frame is the best strategy. It proves that the middle frame does contain more information.
>
> | Strategy |           | With Constraint |           |           | No Constraint |           |
> | :------: | :-------: | :-------------: | :-------: | :-------: | :-----------: | :-------: |
> |          |   R@10    |      R@20       |   R@50    |   R@10    |     R@20      |   R@50    |
> |  First   |   14.38   |      20.45      |   25.60   |   14.85   |     21.73     |   30.68   |
> |   Last   |   14.25   |      20.46      |   25.66   |   14.67   |     21.67     |   30.72   |
> |  Random  |   14.91   |      20.87      |   25.82   |   15.48   |     22.40     |   31.59   |
> |  Middle  | **15.39** |    **21.44**    | **26.24** | **15.83** |   **22.83**   | **31.74** |
>
> **Corresponding Revision**: Sec. 4 (Line 317-324) in the revised paper.

---

> > ### Comment · Reviewer_Ca8D · 2022-12-01
> > **My decision**
> >
> > Thanks for detailed rebuttal. I browsed the responses to my questions and those to all others. Personally, I am impressed by the new weakly-supervised setting for VidSGG. It is very novel for the real applications. The novel approach and detailed ablations can support it is sufficient be accepted.
> > Personally, I would like to raise the final score to 8 based on my experience on cross-media analysis.

---

> > > ### Author Response · Authors · 2022-12-01
> > > **Thanks**
> > >
> > > Thank you for your comments, and we will clarify all the mentioned misunderstandings in the final version.

---

### Official Review · Reviewer_hgye · 2022-10-29

**Confidence:** 3
**Clarity, Quality, Novelty And Reproducibility:** 5. This is a nice engineering work co…
**Correctness:** 2
**Technical Novelty And Significance:** 3
**Empirical Novelty And Significance:** 2
**Recommendation:** 6

**Details Of Ethics Concerns:**

ok to me

**Strength And Weaknesses:**

[advantages]
1. The paper is well-written and easy to follow.
2. The proposed setting is important in practice.
3. The method of using model-based and model-free methods to generate pseudo labels and fusion of the updated weight is interesting. Experiments demonstrate the effectiveness of the proposed PLA.

[weaknesses]
The experiments are somewhat insufficient to justify the claims made by the authors. See suggestions below.

The paper can be improved in the following aspects:

1. The problem setting is claimed to be weakly supervised. However, the proposed method requires a pre-trained object detector. If my understanding is correct, this object detector is trained in a fully supervised setting on ground truth bounding boxes and class labels of a given image. To some extent, the proposed method still requires bounding boxes. It does not fit the weakly supervised setting as described.

2. Follow-up on 1, what if the authors are NOT using pre-trained object detectors or using an imperfect object detector with noisy outputs, would PLA perform efficiently? Can the authors quantify the performance of PLA when there are e.g. 10%, 20%, 40%, and 80% errors made by the pre-trained object detectors?

3. Can the authors show how would the size of the weakly supervised labels influences PLA performance? e.g. the performance of PLA against 10%, 20%, 40%, and 80% annotated video frames out of the entire number of video frames? How would all the baselines perform in these titrated cases?

4. So far, the authors only demonstrate the effectiveness of the method on Action Genome. There is around 1% improvement over various testing scenarios (Table 1). To make the claim convincing, please include the same table but test on another video graph generation dataset to benchmark with SOTA.

5. Data availability (see below)

[minor] Explain what is R (recall) in R@10, R@20 and so on.

**Summary Of The Paper:**

The paper proposed a weakly supervised setting where single-frame ground truth annotation is provided among many video frames; and the single ground truth annotation only provides an unlocalized scene graph. To tackle this problem, a new Psuedo Label Assignment (PLA) method is proposed.

**Summary Of The Review:**

So far, my ranting is weakly rejected due to insufficient experiments; however, I am happy to change the rating depending on the feedback provided by the authors in the rebuttal.

---

> ### Author Response · Authors · 2022-11-18
> **Response to Reviewer hgye (1/3)**
>
> Thank you for the detailed comments. We are willing to address all your questions.
>
> ## Q1: Is still a weakly supervised setting by using pre-trained object detector?
>
> > However, the proposed method requires a pre-trained object detector. ...... To some extent, the proposed method still requires bounding boxes. It does not fit the weakly supervised setting as described.
>
> **A1**: Yes. We explain from two perspectives, the first is about our implementation details and the second is about some previous works.
>
> - We evaluated PLA on the Action Genome dataset, while the pre-trained object detector we use was trained on multiple datasets including COCO, OpenImages, Objects365, and Visual Genome without Action Genome. Our method does not require any bounding boxes in Action Genome, so it is still weakly supervised.
>
> - Previous works[1][2][3] use a similar weakly supervised setting in ImgSGG and also assign pseudo bounding boxes by a pre-trained object detector. They are considered to be weakly supervised methods.
>
> [1] Yiwu Zhong, Jing Shi, Jianwei Yang, Chenliang Xu, and Yin Li. Learning to generate scene graph from natural language supervision. ICCV2021.
>
> [2] Shi J, Zhong Y, Xu N, et al. A simple baseline for weakly-supervised scene graph generation. ICCV2021.
>
> [3] Li X, Chen L, Ma W, et al. Integrating object-aware and interaction-aware knowledge for weakly supervised scene graph generation. ACM MM2022.
>
>
> ## Q2: Imperfect object detector with noisy outputs.
>
> > what if the authors are NOT using pre-trained object detectors or using an imperfect object detector with noisy outputs, would PLA perform efficiently?
>
> **A2**: Thank you for the suggestions. We added some experiments about using an imperfect object detector with noisy outputs.
>
> **Settings**: To simulate an imperfect object detector, we randomly dropped a certain proportion of bounding boxes given by the object detector. Without loss of generality, we set five dropped proportion as 0%, 10%, 20%, 40% and 80%.
>
> **Results**: As shown in Table below, we can observe that the bigger dropped proportion, the worse the performance of PLA, eg. the setting with 0% dropped proportion outperforms settings with 10%, 20%, 40% and 80% dropped proportion by 1.23, 2.21, 3.91 and  10.55, respectively. When dropping 10%, 20% and 40% bboxes, the performance has not decreased much, but the performance decreased significantly if dropping 80% bboxes. Results show that PLA is affected by the preformance of the object detector, but it is the weakness for all two-stage (first detect objects, then predict relations between them) SGG method.
>
> | Dropped Bboxes |           | With Constraint |           |           | No Constraint |           |
> | :------------: | :-------: | :-------------: | :-------: | :-------: | :-----------: | :-------: |
> |                |   R@10    |      R@20       |   R@50    |   R@10    |     R@20      |   R@50    |
> |      0\%       | **15.39** |    **21.44**    | **26.24** | **15.83** |   **22.83**   | **31.74** |
> |      10\%      |   14.16   |      19.86      |   23.60   |   14.66   |     21.12     |   29.48   |
> |      20\%      |   13.18   |      17.85      |   20.99   |   13.66   |     19.46     |   27.26   |
> |      40\%      |   11.48   |      14.86      |   16.53   |   12.01   |     22.14     |   24.72   |
> |      80\%      |   4.84    |      5.29       |   5.32    |   5.62    |     7.15      |   8.48    |
>
> **Corresponding Revision**: Sec. A.5 (Line 560-570) in the revised paper.

---

> > ### Author Response · Authors · 2022-11-18
> > **Response to Reviewer hgye (2/3)**
> >
> > ## Q3: Different size of the weakly supervised labels.
> >
> > > Can the authors show how would the size of the weakly supervised labels influences PLA performance?
> >
> > **A3**: Thank you for the suggestions. We added some experiments about different size of the weakly supervised labels.
> >
> > **Settings**: Although PLA is designed for single-frame weak supervision, it can be easily extended to multi-frame weak supervision. For the model-based teacher, we just train the model with more frames. For the model-free teacher, we take each annotated frame as a starting point to propagate relations. Without loss of generality, we set four annotated proportion as 10%, 20%, 40% and 80%. We annotated frame evenly in temporal axis, for example, if a 20-frame video with 20% annotated proportion, we will annotate its 1st, 7th, 14th and 20th frame.
> >
> > **Results**: As shown in Table below, we can observe that the more annotated frames, the better the performance of PLA, e.g., model with 80% annotation frames outperforms models with 10%, 20% and 40% annotation frames 1.16, 0.26 and 0.03. It proves that PLA can easily adapt to multi-frame weak supervision setting.
> >
> > | Proportion |           | With Constraint |           |           | No Constraint |           |
> > | :--------: | :-------: | :-------------: | :-------: | :-------: | :-----------: | :-------: |
> > |            |   R@10    |      R@20       |   R@50    |   R@10    |     R@20      |   R@50    |
> > |  1 frame   |   15.39   |      21.44      |   26.24   |   15.83   |     22.83     |   31.74   |
> > |    10\%    |   15.10   |      21.17      |   25.96   |   15.55   |     22.56     |   31.88   |
> > |    20\%    |   16.00   |      21.91      |   26.61   |   16.56   |     23.55     |   32.89   |
> > |    40\%    |   16.23   |      22.50      |   27.37   |   16.83   |     24.07     | **33.30** |
> > |    80\%    | **16.26** |    **22.57**    | **27.68** | **16.89** |   **24.18**   |   33.26   |
> >
> > **Corresponding Revision**: Sec. A.6 (Line 571-581) in the revised paper.
> >
> >
> > ## Q4: Result on other video graph generation dataset.
> >
> > > To make the claim convincing, please include the same table but test on another video graph generation dataset to benchmark with SOTA.
> >
> > **A4**: So far, the frame-level VidSGG task only has one dataset - Action Genome. Existing studies [4][5][6] on frame-level VidSGG only evaluate their methods on AG dataset. Therefore, we can not evaluate SF-VidSGG task on other dataset yet.
> > According for your comments, we do another experiment to show PLA is general.
> > Setting: We used DSG-DETR as the student model of PLA. Since PLA is agnostic to the specific VidSGG architecture, it can be easily incorporated into any advanced VidSGG models. We used the same teachers as for STTran to generate pseudo labels for DSG-DETR, then trained a fully-supervised VidSGG model by these pseudo labels.
> > Results: As shown in Table 2, PLA with DSG-DETR also achieves similar performance with STTran across all the metrics. These results show that PLA is a general framework and can easily extend to advanced fully supervised VidSGG methods.
> >
> > | Supervision |  Model   |   PLA    |          | With Constraint |          |          | No Constraint |          |
> > | :---------: | :------: | :------: | :------: | :-------------: | :------: | :------: | :-----------: | :------: |
> > |             |          |          |   R@10   |      R@20       |   R@50   |   R@10   |     R@20      |   R@50   |
> > |    Weak     |  STTran  | &#10004; |   15.4   |    **21.4**     | **26.2** |   15.8   |   **22.8**    |   31.7   |
> > |    Weak     | DSG-DETR | &#10004; | **15.5** |      21.3       |   25.9   | **15.9** |     22.7      | **31.9** |
> >
> > **Corresponding Revision**: Sec. 4 (Line 291-298) in the revised paper.
> >
> > [4] Cong Y, Liao W, Ackermann H, et al. Spatial-temporal transformer for dynamic scene graph generation.ICCV2021.
> >
> > [5] Li Y, Yang X, Xu C. Dynamic Scene Graph Generation via Anticipatory Pre-Training. CVPR2022.
> >
> > [6] Feng S, Tripathi S, Mostafa H, et al. Exploiting Long-Term Dependencies for Generating Dynamic Scene Graphs. WACV2023.
> >
> >
> > ## Q5: What is R(recall) in R@10, R@20?
> >
> > > Explain what is R (recall) in R@10, R@20 and so on.
> >
> > **A5**: Recall is the most widely used metric in the scene graph generation(SGG) task. **Recall@K** computes the fraction of times the correct relationship is predicted in the top **K** confident relationship predictions. In SGG task, annotators can not exhaustively annotate all possible relationships in an image [7], so we choose recall instead of precision as the evaluation metric. Specifically, models may predict correct relations which are not annotated in the ground truth, then precision will penalize the predictions but recall will not.
> >
> > [7] Cewu Lu, Ranjay Krishna, Michael Bernstein, and Li Fei-Fei. Visual relationship detection with language priors. In European conference on computer vision, pp. 852–869. Springer, 2016.

---

> > > ### Author Response · Authors · 2022-11-18
> > > **Response to Reviewer hgye (3/3)**
> > >
> > > ## Q6: Reproducibility.
> > >
> > > > Given the complicated designs of various components and hyperparameters, would the source code
> > > become available for reproducibility purposes? Please include such statements in the paper.
> > >
> > > **A6**: We guarantee the reproducibility of our method from two aspects.
> > >
> > > - We have already put the reproducibility statement on Line 361-364 in the origin paper. We will release the code in the future.
> > > - We provided more detailed implementation details in the Appendix to ensure all experiments can be reproduced.
> > >
> > > **Corresponding Revision**: Sec. reproducibility statement (Line 348-351) in the revised paper.

---

> > > > ### Comment · Reviewer_hgye · 2022-11-24
> > > > **thank you**
> > > >
> > > > The authors have clarified most of my questions. I appreciated the efforts and clarification!
> > > > However, I am still not convinced because:
> > > > 1. In A2 and A3 answers from the authors, can the authors show similar tables for other baselines as well? The reasoning is to assess how efficient the method is in various weakly-supervised settings.
> > > > 2.  In A4, I am observing marginal performance improvements. I am a bit concerned about whether the proposed method is effective or not.
> > > >
> > > > I would like to keep my rating the same.

---

> > > > > ### Author Response · Authors · 2022-11-27
> > > > > **Answer to reviewer hgye**
> > > > >
> > > > > We thank the reviewer for the reply, and we are happy to explain our work more clearly.
> > > > >
> > > > > - PLA is a model-agnostic method. PLA has two steps: first to assign pseudo labels for the video, then to train a fully-supervised VidSGG model by these pseudo labels. Therefore, PLA is agnostic to different VidSGG architectures, it can be easily incorporated into any advanced VidSGG model.
> > > > > - However, to the best of our knowledge,there are only three existing fully-supervised methods: STTran [1], APT [2] and DSG-DETR [3]. Besides, APT has not released its code yet, so we only evaluate PLA with STTran and DSG-DETR.
> > > > > - The table we drew before may be confusing, so we redrew the table for Q4. From the results in the Table below, we have the following observations: 1) The fully supervised baseline models (#1 and #4) cannot apply to the SF-VidSGG task directly due to the lack of bounding box annotations. The pseudo labels for bounding boxes assigned by Obj_PLA are necessary for model training. In our previous replies, we took models with Obj_PLA (#2 and #5) as our baselines. 2) Compared to models only with Obj_PLA (#2 and #5), models with all three components of PLA (#3 and #6) can achieve better performance on all R@K metrics, i.e., in terms of R@10 with constraint criteria, #3 improves by 7.47% (15.39 v.s.14.32) relatively over #2, and #6 improves by 10.54% (15.42 v.s.13.95) relatively over #5. Thus, PLA is effective for the SF-VidSGG task.
> > > > >
> > > > > | Supervision |   #   |         Model          |         | With Constraint |         |         | No Constraint |         |
> > > > > | :---------: | :---: | :--------------------: | :-----: | :-------------: | :-----: | :-----: | :-----------: | :-----: |
> > > > > |             |       |                        |  R@10   |      R@20       |  R@50   |  R@10   |     R@20      |  R@50   |
> > > > > |             |   1   |         STTran         |    -    |        -        |    -    |    -    |       -       |    -    |
> > > > > |    Weak     |   2   |  STTran$_{+Obj-PLA}$  |  14.32  |      20.42      |  25.43  |  14.78  |     21.72     |  30.87  |
> > > > > |             |   3   |    STTran$_{+PLA}$     | $15.39$ |     $21.44$     | $26.24$ | $15.83$ |    $22.83$    | $31.74$ |
> > > > > |             |   4   |        DSG-DETR        |    -    |        -        |    -    |    -    |       -       |    -    |
> > > > > |    Weak     |   5   | DSG-DETR$_{+Obj-PLA}$ |  13.95  |      20.08      |  25.10  |  14.29  |     21.25     |  31.08  |
> > > > > |             |   6   |   DSG-DETR$_{+PLA}$    | $15.42$ |     $21.28$     | $25.89$ | $15.87$ |    $22.73$    | $31.86$ |
> > > > >
> > > > > - About Q2 and Q3, we can evaluate PLA with DSG-DETR under these various weakly-supervised settings but it may time-consuming. From our point of view, the final conclusions should be the same as A2/A3. For Q2, the object detector is an important part of existing SGG methods, so the final performance will drop with an imperfect object detector. For Q3, with more labeled frames, PLA can assign more accurate pseudo labels, so the final performance will improve with more weakly supervised labels.
> > > > >
> > > > > We hope our responses have answered your questions. We are happy to answer any questions you may have later.
> > > > >
> > > > > [1] Cong Y, Liao W, Ackermann H, et al. Spatial-temporal transformer for dynamic scene graph generation. ICCV2021.
> > > > >
> > > > > [2] Li Y, Yang X, Xu C. Dynamic Scene Graph Generation via Anticipatory Pre-Training. CVPR2022.
> > > > >
> > > > > [3] Feng S, Tripathi S, Mostafa H, et al. Exploiting Long-Term Dependencies for Generating Dynamic Scene Graphs. arXiv:2112.09828, 2021.

---

> > > > > > ### Comment · Reviewer_hgye · 2022-11-28
> > > > > > **thank you - 2**
> > > > > >
> > > > > > Thank you for answering my questions.
> > > > > > As other reviewers pointed out, the work is somewhat engineering and incremental. However, I think that this is still an interesting paper that takes many aspects into account and integrates various components into one framework. The model yields marginally better performance.
> > > > > >
> > > > > > I have revised my rating.

---

> > > > > > > ### Author Response · Authors · 2022-11-28
> > > > > > > **thank you**
> > > > > > >
> > > > > > > We thank the Reviewer for all their relevant and constructive criticism to improve the quality of our work.

---

### Official Review · Reviewer_XWwS · 2022-10-31

**Confidence:** 3
**Correctness:** 3
**Technical Novelty And Significance:** 2
**Empirical Novelty And Significance:** 2
**Recommendation:** 6

**Clarity, Quality, Novelty And Reproducibility:**

Quality: I think the approach is not novel and experiments are not sufficient.
Orginality, Clarity: Good. Easy to follow.
Originality: The main contribution is to use weak supervision for VidSGG task, but the proposed method is not very novel.

**Strength And Weaknesses:**

Strength:
1. The paper propose to do vidSGG with weak supervision, which is a good contribution.
2. The proposed approach of pseudo label is reasonable and easy to follow.
3. The ablation study shows the effectiveness of proposed modules.

Weakness:
1. Approach

a. The approach seems only supports short video without drastic changes. If the scene changes, the proposed approach may not work since the pseudo label assigiment will not be correct.

b. The approach is not very novel. The pipeline is a simple combination of three parts, and all of them are not very new, but just a simple pseudo label->training pipeline.

2. Experiments

a. How to choose the annotation frame? Is it chosen randomly, or mid-one? How about choose the first frame? An experiment of different annotation frame choosing strategy should be added.

b. Should add experiments to compare with fully-supervised approach to show how much is behind this upper-bound.

c. Should compare with existing VidSGG methods metioned in Related Work.

d. Please show more qualitative illustrations. Maybe in supplementary file.


**Summary Of The Paper:**

The paper propose a new task that only leverage a single frame with unlocalized annoation for video-level scene graph generation. To do this, the author propose an approach called PLA to create pseudo labels for unannotated frames, and train the fully-supervised VidSGG model with these pseudo labels.  The results on Action Genome benchmark demonstrates the effectiveness of PLA.

**Summary Of The Review:**

Due to the lack of experiments and the lack of novelty, I think the paper is below the acceptance bar.

---

> ### Author Response · Authors · 2022-11-18
> **Response to Reviewer XWwS (1/3)**
>
> Thank you for the detailed comments. We are willing to address all your questions.
>
> ## Q1: The approach seems only supports short video without drastic changes?
>
> > If the scene changes, the proposed approach may not work since the pseudo label assigiment will not be correct.
>
> **A1**: We explain this problem from two aspects.
>
> - Our method designed two different teachers to assign pseudo labels. The model-based teacher is not affected by drastic changes because their training data is ground truth relation. Specifically, they can handle different situations: the model-free teacher performs better for videos with less dynamic relationships, while the model-based teacher performs better for videos with drastic changes.
>
> - The future predicate prediction module assigns adapted weights of the pseudo labels by relation transition and teacher makes wrong prediction for drastic changes will be given low weights. Therefore, PLA is less affected by the drastic changes.
>
> ## Q2: The method is not novel.
>
> > The pipeline is a simple combination of three parts, and all of them are not very new, but just a simple pseudo label->training pipeline.
>
> **A2**: Thanks for recognizing that the new proposed task is a good contribution and our proposed approach of pseudo label is reasonable and easy to follow. As for the novelty of this proposed method, we would like to make several clarifications:
>
> 1. **A new framework for the new task**: At first, we acknowledge that knowledge distillation or data augmentation are widely-used approaches in many different computer vision or deep learning based models. But we think how to effectively design a framework by reasonable improvements to solve a new but meaningful task (ie, SF-VidSGG) itself is also novel.
>
> 2. **Two simple but effective teachers**: Although multi-teacher knowledge distillation (KD) is an existing technique, one of the key differences in different KD works are the designs or choices of different teachers. In this work, we design two simple but effective teachers for SF-VidSGG task. The model-based teacher is focused on the spatial context and the model-free teacher is focused on the temporal context. We can dynamically fuse these two complement teachers to get better pseudo labels. Besides, PLA is a general method and it can be easily extended to advanced teachers or more teachers.
>
> 3. **A future predicate prediction module**: it leverages relation transition to assign adapted weights. Relation transiation is one of the key characteristics of the visual relations in the videos, ie, modeling the relation transition is quite a straightforward idea in Video SGG. Compared to existing work [1], we have two main differences:
>
>     - modeling of relation transition is the final target of [1], while we modeled it for a better pseudo label assignment.
>     - [1] designed a complex model based on transformer to model relation transition, but PLA's submodel to predict relation transition is very simple which only contains a matrix T(Line 226-227 in the revised paper).
>
>     Therefore, one of our contributions is to utilize relation transition to assign better pseudo labels, not to model relation transition.
>
> [1] Li Mi, Yangjun Ou, and Zhenzhong Chen. Visual relationship forecasting in videos. arXiv preprint arXiv:2107.01181, 2021.

---

> > ### Author Response · Authors · 2022-11-18
> > **Response to Reviewer XWwS (2/3)**
> >
> > ## Q3: How to choose the annotation frame?
> >
> > > An experiment of different annotation frame choosing strategy should be added.
> >
> > **A3**: In our paper, we choose the middle frame as the annotaion frame. First, to minimize labeling costs, we only annotate one frame in each frame. Second, since the main event of the video is usually in the middle, we think model trained with middle-annotated frame has the best performance.
> >
> > Following your suggestions, we added experiments of different annotation frame choosing strategy for the sake of thoroughness.
> >
> > **Settings**: We designed four simple annotation frame choosing strategy. "First" means annotating the first frame with an unlocalized scene graph in each video, "Last" means choosing the last frame, "Random" means choosing one frame randomly, "Middle" means choosing the middle frame.
> >
> > **Results**: As results shown in Table below, We can observe that model trained by the middle annotation frame has the best performance, eg., in terms of R@10 with constraint criteria for SGDet, strategy "Middle" outperforms "First", "Last" and "Random" by 1.01, 1.14 and 0.48, respectively. So the conclusion is annotating the middle frame is the best strategy. It proves that the middle frame does contain more information.
> >
> > | Strategy |           | With Constraint |           |           | No Constraint |           |
> > | :------: | :-------: | :-------------: | :-------: | :-------: | :-----------: | :-------: |
> > |          |   R@10    |      R@20       |   R@50    |   R@10    |     R@20      |   R@50    |
> > |  First   |   14.38   |      20.45      |   25.60   |   14.85   |     21.73     |   30.68   |
> > |   Last   |   14.25   |      20.46      |   25.66   |   14.67   |     21.67     |   30.72   |
> > |  Random  |   14.91   |      20.87      |   25.82   |   15.48   |     22.40     |   31.59   |
> > |  Middle  | **15.39** |    **21.44**    | **26.24** | **15.83** |   **22.83**   | **31.74** |
> >
> > **Corresponding Revision**: Sec. 4 (Line 317-324) in the revised paper.
> >
> >
> > ## Q4: More comparisons with fully-supervised approach.
> >
> > > Should add experiments to compare with fully-supervised approach to show how much is behind this upper-bound.
> >
> > **A4**: There are three existing fully-supervised method: STTran [2], APT [3] and DSG-DETR [4]. We added comparisons with these three fully-supervised approaches.
> >
> > **Results**: As shown in Table below, we can observe that there is still a gap between the performance of PLA and other fully-supervised approaches (e.g., 15.4 vs. 25.2 for STTran in terms of R@10 with constraint criteria for SGDet). However, considering that the annotation cost of our method is extremely lower (<5%) than that of full supervision. we think it is still a good start for weakly supervised frame-level VidSGG. We leave the task to reduce the gap as the future work.
> >
> > | Supervision |    Model    |   PLA    |          | With Constraint |          |          | No Constraint |          |
> > | :---------: | :---------: | :------: | :------: | :-------------: | :------: | :------: | :-----------: | :------: |
> > |             |             |          |   R@10   |      R@20       |   R@50   |   R@10   |     R@20      |   R@50   |
> > |    Weak     |  STTran[2]  | &#10004; |   15.4   |      21.4       |   26.2   |   15.8   |     22.8      |   31.7   |
> > |    Full     |  STTran[2]  | &#10008; |   25.2   |      34.1       |   37.0   |   24.6   |     36.2      |   48.8   |
> > |    Full     |   APT[3]    | &#10008; |   26.3   |    **36.1**     | **38.3** |   25.7   |     37.9      | **50.1** |
> > |    Full     | DSG-DETR[4] | &#10008; | **30.3** |      34.8       |   36.1   | **32.1** |   **40.9**    |   48.3   |
> >
> > **Corresponding Revision**: Sec. 4 (Line 283-290) in the revised paper.
> >
> > [2] Cong Y, Liao W, Ackermann H, et al. Spatial-temporal transformer for dynamic scene graph generation. ICCV2021.
> >
> > [3] Li Y, Yang X, Xu C. Dynamic Scene Graph Generation via Anticipatory Pre-Training. CVPR2022.
> >
> > [4] Feng S, Tripathi S, Mostafa H, et al. Exploiting Long-Term Dependencies for Generating Dynamic Scene Graphs. arXiv:2112.09828, 2021.

---

> > > ### Author Response · Authors · 2022-11-18
> > > **Response to Reviewer XWwS (3/3)**
> > >
> > > ## Q5: Comparison with existing VidSGG methods.
> > >
> > > > Should compare with existing VidSGG methods metioned in Related Work.
> > >
> > > **A5**: There are three existing fully-supervised methods: STTran [2], APT [3], and DSG-DETR [4]. We already apply PLA to the STTran method. However, APT has not released its code yet, so it is time-consuming for us to apply PLA to APT. DSG-DETR utilizes object tracking to capture long-term dependencies, which means it is difficult to achieve good performance if only one frame for each video is given (like model A in Table 1).
> > >
> > > **Setting**: Since we can not simply modify DSG-DETR to an ImgSGG model without a large performance loss, we use model A as the model-based teacher again. Then we use the same setting as for STTran to generate pseudo labels for DSG-DETR, then trained a fully-supervised VidSGG model by these pseudo labels.
> > >
> > > **Results**: As shown in Table below, PLA with DSG-DETR also achieves similar performance with STTran across all the metrics. These results show that PLA is a general framework and can easily extend to advanced fully supervised VidSGG methods.
> > >
> > > | Supervision |  Model   |   PLA    |          | With Constraint |          |          | No Constraint |          |
> > > | :---------: | :------: | :------: | :------: | :-------------: | :------: | :------: | :-----------: | :------: |
> > > |             |          |          |   R@10   |      R@20       |   R@50   |   R@10   |     R@20      |   R@50   |
> > > |    Weak     |  STTran  | &#10004; |   15.4   |    **21.4**     | **26.2** |   15.8   |   **22.8**    |   31.7   |
> > > |    Weak     | DSG-DETR | &#10004; | **15.5** |      21.3       |   25.9   | **15.9** |     22.7      | **31.9** |
> > >
> > > **Corresponding Revision**: Sec. 4 (Line 291-298) in the revised paper.
> > >
> > > ## Q6: Qualitative illustrations.
> > >
> > > > Please show more qualitative illustrations. Maybe in supplementary file.
> > >
> > > **A6**: We added more the qualitative illustrations in the supplementary file. Results show observe that PLA has the ability to detect dynamic relations.
> > >
> > > **Corresponding Revision**: Sec. A.7 (Line 582-585) in the revised paper.

---

> > > > ### Comment · Reviewer_XWwS · 2022-11-28
> > > > **Thanks for authors' feedback**
> > > >
> > > > Thanks for authors' response, and most of my questions have been answered.
> > > >
> > > > However, my major concern is the performance:
> > > >
> > > > Weakly baseline: 14.3, proposed approach: 15.4, fully-supervised approach: 25.2 or 30.3.
> > > >
> > > > Considering the fully-supervised upper bound (there is still a 10 or 15% gap between the proposed approach and fully-supervised approach), I think the performance is too close to the baseline and the improvement is not significant (only 1%), indicating that the proposed approach is not very effective.
> > > >
> > > > Also, considering the approach is not very novel, I would like to keep my rating.

---

> > > > > ### Author Response · Authors · 2022-11-28
> > > > > **Answer to reviewer XWwS**
> > > > >
> > > > >
> > > > > We thank the reviewer for the reply, and we are happy to explain our work more clearly.
> > > > >
> > > > > ## About the gap between PLA and the fully-supervised upper bound.
> > > > >
> > > > > PLA is a weakly supervised method for the VidSGG task, so it is unfair to compare PLA with fully supervised methods. We can consider the gap between weakly supervised and fully supervised methods on other scene understanding tasks, such as object detection, another task in computer vision. In 2016, the best weakly supervised object detection method was WSDDN [1], while the best fully supervised method was SSD [2]. On PASCAL VOC 2007 dataset, the results of WSDDN and SSD are **39.3%** and **81.6%** mAP, respectively. Although there is a **42.3%** gap, WSDDN is still regarded as a pioneering work in this field. Until now, the gap between weakly supervised methods and fully supervised methods is still significant. On MS COCO 2017 dataset, the SOTA weakly supervised method, SOS [3], reaches **32.8%** mAP_50, while the SOTA fully supervised method, Co-DETR [4], reaches **77.4%** mAP_50. There is still a **44.6%** gap between the weakly supervised approach and fully supervised approach on the object detection task.
> > > > >
> > > > > In our opinion, the big gap between weakly and fully supervised methods shows that SF-VidSGG is a challenging task. The gap with fully supervised methods does not represent the weakness of weakly supervised methods.
> > > > >
> > > > > ## About the performance improvement between PLA and the baseline.
> > > > >
> > > > > The table we drew before may be confusing, so we redrew the table to show the effectiveness and generalization of PLA more clearly. From the results in the Table below, we have the following observations:
> > > > >
> > > > > - **Baselines**: In our previous replies, we took models with Obj_PLA (#2 and #5) as our baselines because the fully supervised baseline models (#1 and #4) cannot apply to the SF-VidSGG task directly due to the lack of bounding box annotations. The pseudo labels for bounding boxes assigned by Obj_PLA are necessary for model training, which means PLA has already made contributions to our previous baselines (#2 and #5). Therefore,  when discussing the effectiveness of our model, we should not only consider the comparison between #2 and #3 but also consider the comparison between #1 and #3.
> > > > >
> > > > > - **Effectiveness**: Compared to models only with Obj_PLA (#2 and #5), models with all three components of PLA (#3 and #6) can achieve better performance on all R@K metrics, i.e., in terms of R@10 with constraint criteria, #3 improves by 7.47% (15.39 v.s.14.32) relatively over #2, and #6 improves by 10.54% (15.42 v.s.13.95) relatively over #5. Thus, PLA is effective for the SF-VidSGG task.
> > > > >
> > > > > - **Generalization**: Since PLA is agnostic to the specific VidSGG architecture, it can be easily incorporated into any advanced VidSGG model. We evaluate PLA with STTran [5] and DSG-DETR [6]. As shown in Table below, PLA with DSG-DETR (#6) also achieves similar performance with STTran (#3) across all the metrics. These results show that PLA is a general framework.
> > > > >
> > > > > | Supervision |   #   |         Model          |         | With Constraint |         |         | No Constraint |         |
> > > > > | :---------: | :---: | :--------------------: | :-----: | :-------------: | :-----: | :-----: | :-----------: | :-----: |
> > > > > |             |       |                        |  R@10   |      R@20       |  R@50   |  R@10   |     R@20      |  R@50   |
> > > > > |             |   1   |         STTran         |    -    |        -        |    -    |    -    |       -       |    -    |
> > > > > |    Weak     |   2   |  STTran$_{+Obj-PLA}$  |  14.32  |      20.42      |  25.43  |  14.78  |     21.72     |  30.87  |
> > > > > |             |   3   |    STTran$_{+PLA}$     | $15.39$ |     $21.44$     | $26.24$ | $15.83$ |    $22.83$    | $31.74$ |
> > > > > |             |   4   |        DSG-DETR        |    -    |        -        |    -    |    -    |       -       |    -    |
> > > > > |    Weak     |   5   | DSG-DETR$_{+Obj-PLA}$ |  13.95  |      20.08      |  25.10  |  14.29  |     21.25     |  31.08  |
> > > > > |             |   6   |   DSG-DETR$_{+PLA}$    | $15.42$ |     $21.28$     | $25.89$ | $15.87$ |    $22.73$    | $31.86$ |
> > > > >
> > > > >
> > > > > We hope our responses have answered your questions. We are happy to answer any questions you may have later.
> > > > >
> > > > > [1] Bilen H, Vedaldi A. Weakly Supervised Deep Detection Networks. CVPR2016.
> > > > >
> > > > > [2] Liu W, Anguelov D, Erhan D, et al. SSD: Single Shot MultiBox Detector. ECCV2016.
> > > > >
> > > > > [3] Sui L, Zhang C L, Wu J. Salvage of Supervisionin Weakly Supervised Object Detection. CVPR2022.
> > > > >
> > > > > [4] Zong Z, Song G, Liu Y. DETRs with Collaborative Hybrid Assignments Training. arXiv preprint arXiv:2211.12860, 2022.
> > > > >
> > > > > [5] Cong Y, Liao W, Ackermann H, et al. Spatial-temporal transformer for dynamic scene graph generation. ICCV2021.
> > > > >
> > > > > [6] Feng S, Tripathi S, Mostafa H, et al. Exploiting Long-Term Dependencies for Generating Dynamic Scene Graphs. arXiv:2112.09828, 2021.

---

> > > > > > ### Comment · Reviewer_XWwS · 2022-11-30
> > > > > > **Thanks for authors' feedback**
> > > > > >
> > > > > > Thanks for authors' thorough feedback!
> > > > > >
> > > > > > 1. from 14.3 to 15.4: I understand that 14.3 is also a result of the proposed method, so this question is well anwered.
> > > > > >
> > > > > > 2. Performance gap. For [1,3] and [2,4], [1,3] has only image-level label without any location knowledge, so it is very challenging. But in this work, the author seems to adopt a pretrained detector, which provides location information (maybe  this detector utilized some bounding-box annotation in the its own dataset? If so, I think this work may indirectly utilize the bounding-box annotation in other dataset. Is my understanding correct?). Hence, the pesudo label assignment in this task is not very challenging, so I think the performance gap to the fully-supervised approach is still quite large.

---

> > > > > > > ### Author Response · Authors · 2022-11-30
> > > > > > > **Clarification for the object detector**
> > > > > > >
> > > > > > > Thank you for your questions.  For the question about the pretrained object detector, we could explain from three perspectives:
> > > > > > >
> > > > > > > + **Implementation Details**: We evaluated PLA on the Action Genome dataset, while the pre-trained object detector we use was trained on multiple datasets including COCO, OpenImages, Objects365, and Visual Genome without Action Genome. Our method does not require any bounding boxes in Action Genome, so it is still weakly supervised.
> > > > > > >
> > > > > > > + **Same Convention in Previous Weakly-Supervised Works**: Similarly, almost all previous weakly-supervised image scene graph generation works[1][2][3] also use a similar weakly supervised setting. Specifically, they assign pseudo bounding boxes by a pre-trained object detector, where the scene graph generation model is evaluated on Visual Genome, and the object detector is trained on other datasets (eg, OpenImages). And all of them are still be considered as weakly supervised methods.
> > > > > > >
> > > > > > > + **Performance Gaps**: As for the performance gaps between the weakly-supervised setting and fully-supervised setting, the performance gaps in image scene graph generation are even much higher. For the state-of-the-art fully-supervised model (in year 2018) [4] achieves 25.48% at R@20 on SGDet, while the recent weakly-supervised models (in year 2021 and 2022) [1][2][3] achieve 4.12%, 7.23% and 9.57% (vs. 25.48% in year 2018), respectively.
> > > > > > >
> > > > > > > [1] Yiwu Zhong, et al. Learning to generate scene graph from natural language supervision. ICCV2021.
> > > > > > >
> > > > > > > [2] Jing Shi, et al. A simple baseline for weakly-supervised scene graph generation. ICCV2021.
> > > > > > >
> > > > > > > [3] Xingchen Li, et al. Integrating object-aware and interaction-aware knowledge for weakly supervised scene graph generation. ACM MM2022.
> > > > > > >
> > > > > > > [4] Rowan Zellers, et al. Neural motifs: Scene graph parsing with global context. In CVPR, 2018.
> > > > > > >
> > > > > > > We are happy to address all your concerns, and please let us know if you have any other questions.

---

> > > > > > > > ### Comment · Reviewer_XWwS · 2022-12-01
> > > > > > > > **Thanks for author's feedback**
> > > > > > > >
> > > > > > > > Thanks for author's feedback. I think despite that there is still a certain performance gap, the paper is above the acceptance threshold, due to the new task, the reasonable approach, good writing and presentation.
> > > > > > > >
> > > > > > > > I have raised the score to 6. Thanks.

---

> > > > > > > > > ### Author Response · Authors · 2022-12-01
> > > > > > > > > **Thanks**
> > > > > > > > >
> > > > > > > > > Thank you so much for raising your rating to positive, and we will clarify all the misunderstandings in the revision.

---

### Official Review · Reviewer_spBp · 2022-11-02

**Confidence:** 4
**Correctness:** 3
**Technical Novelty And Significance:** 3
**Empirical Novelty And Significance:** 2
**Recommendation:** 6

**Clarity, Quality, Novelty And Reproducibility:**

The paper is globally well written with some minor rough edges in the text. The proposed task is interesting yet the solution is less innovative. For reproducibility, some implementation details are provided in the paper and the authors promised to release the code in the future.

**Strength And Weaknesses:**

**Strength**

In spite of an extension from images to videos, the proposed task of weakly supervised video scene graph generation is new.

The proposed method harnesses multiple cues (object detector, image scene graph model, transition probability between predicates) to generate pseudo scene graph labels for training. The method is interesting and could provide a baseline for future research.

**Weakness**

While the task and the method is interesting, the technical components of the proposed method are less exciting. Many of the ideas can be found in prior works, including (1) using object detector to ground unlocalized scene graphs (e.g., Zhong et al., 2021), (2) distilling from another model (see works on knowledge distillation), (3) using visual tracking to augment training samples (described as model-free teacher), and (4) modeling of relation transitions (Mi et al., 2021).  The proposed method seems like a reasonable combination of multiple existing ideas from video understanding and image scene graph generation. It is thus difficult to gauge the key innovation here.

The gap between a strawman baseline (learning a model from the same set of sparse frames with unlocalized scene graphs) and the proposed method is rather small (1% on SGDet). In fact, a stronger and perhaps more realistic baseline is listed as model C in Table 1 (within a gap of 0.7% on SGDet), where a model is trained using sparse scene graphs plus augmented annotations from simple tracking. It is not totally clear if this gap is sufficient to justify the proposed method, which is arguably more sophisticated in its training scheme.

**Other Minor Comments**

A key argument in the introduction to motivate this work is that relations that involve motion (e.g., walking vs. running) can not be readily distinguished in static images. This unfortunately remains an issue for the proposed method, as it relies on image features from an object detector (L285). I wonder if the authors have considered using video features (e.g., from 3D convolutional networks or video transformers) for the proposed task.

I thought the baseline listed Table 2 is referring to model A in Table 1, yet the numbers do not quite match. It will be great if the authors can provide some clarification here.

L322, I think the text is referring to Table 2 instead of Table 1.


**Summary Of The Paper:**

The paper extended weakly supervised image scene graph generation to videos, and presented a new task of weakly supervised video scene graph generation, where a model must learn from unlocalized scene graphs on a sparse set of video frames. In tandem with the new task, the paper also proposed an interesting solution. The key to the solution is to generate pseudo scene graph labels by leveraging existing object detectors, distilling from image scene graph models, and modeling temporal continuity in videos. The proposed method is evaluated on the Action Genome dataset with some promising results.


**Summary Of The Review:**

The paper presented an interesting new task of weakly supervised video scene graph generation, and proposed a solution to the task. While the solution is quite reasonable and combines several existing ideas, it lacks technical innovation and the results are less satisfactory. Overall, I am not enthusiastic about this paper.

The author response has addressed most of concerns, and I have raised my rating accordingly.

---

> ### Author Response · Authors · 2022-11-18
> **Response to Reviewer spBp (1/3)**
>
> Thank you for the detailed comments. We are willing to address all your questions.
>
> ## Q1: The solution is less innovate.
>
> > While the task and the method is interesting, the technical compoents of the prompsed method are less exciting. Many of the ideas can be found in prior works. ...... The proposed method seems like a reasonable combination of multiple existing ideas from video understanding and image scene graph generation.
>
> **A1**: Thanks for recognizing that the new proposed task and method are interesting, ie, **the contribution of the paper is not only the specific PLA method but also the first weakly-supervised Video SGG task setting**. As for the novelty of this proposed method, we would like to make several clarifications:
>
> 1. **A new framework for the new task**: At first, we acknowledge that knowledge distillation or data augmentation are widely-used approaches in many different computer vision or deep learning based models. But we think how to effectively design a framework by reasonable improvements to solve a new but meaningful task (ie, SF-VidSGG) itself is also novel.
>
> 2. **Two simple but effective teachers**: Although multi-teacher knowledge distillation (KD) is an existing technique, one of the key differences in different KD works are the designs or choices of different teachers. In this work, we design two simple but effective teachers for SF-VidSGG task. The model-based teacher is focused on the spatial context and the model-free teacher is focused on the temporal context. We can dynamically fuse these two complement teachers to get better pseudo labels. Besides, PLA is a general method and it can be easily extended to advanced teachers or more teachers.
>
> 3. **A future predicate prediction module**: it leverages relation transition to assign adapted weights. Relation transiation is one of the key characteristics of the visual relations in the videos, ie, modeling the relation transition is quite a straightforward idea in Video SGG. Compared to existing work [1], we have two main differences:
>
>     - modeling of relation transition is the final target of [1], while we modeled it for a better pseudo label assignment.
>     - [1] designed a complex model based on transformer to model relation transition, but PLA's submodel to predict relation transition is very simple which only contains a matrix T(Line 226-227 in the revised paper).
>
>     Therefore, one of our contributions is to utilize relation transition to assign better pseudo labels, not to model relation transition.
>
> [1] Li Mi, Yangjun Ou, and Zhenzhong Chen. Visual relationship forecasting in videos. arXiv preprint arXiv:2107.01181, 2021.

---

> > ### Author Response · Authors · 2022-11-18
> > **Response to Reviewer spBp (2/3)**
> >
> > ## Q2: The gap between PLA and baseline is insufficient.
> >
> > > It is not totally clear if this gap is sufficient to justify the proposed method, which is arguably more sophisticated in its training scheme.
> >
> > **A2**: We explain this problem from two aspects: First is about comparing with upper-bound performance. we can find that PLA can reduce the performance gap between the baseline and the upper-bound. The second is about the generalization of PLA. We can apply PLA to other VidSGG models and achieve similar performance.
> >
> > ### **Comparison with upper-bound performance**
> >
> > Like most existing SGG methods, PLA has two main steps: the first is to detect objects in the image, and the second is to predict relations between these objects. The performance of PLA is mainly affected by the first step due to the poor performance of the off-shelf detector we used. Under this object detection strategy, the upper-bound performance is not very high, but PLA can effectively reduce the performance gap between the baseline and the upper-bound.
> >
> > **Settings**: An upper-bound model can be trained as follows: first detect objects by Obj-PLA, then assign ground truth relation labels between them, and finally use these labels to train the student model. We named this upper-bound model as GTRel. Results are shown in the table below. Note that the fully supervised baseline models (#1) cannot apply to the SF-VidSGG task directly due to the lack of bounding box annotations, so our baseline model is STTran with Obj-PLA (#2).
> >
> > **Results**: As results shown in the Table below, we demonstrate that model with PLA (#3) can reduce the gap between the baseline model (#2) and GTRel (#4). For example, #4 improves by 12.43% (16.10 v.s.14.32) relatively over #2 in terms of R@10 with constraint criteria for SGDet, but only improves by 4.61% (16.10 v.s.15.39) relatively over #3 on the same metric. These results show that PLA can reduce more than half the performance gap between the baseline and the upper-bound for the SF-VidSGG task.
> >
> > | Supervision |   #   |        Model        |           | With Constraint |           |           | No Constraint |           |
> > | :---------: | :---: | :-----------------: | :-------: | :-------------: | :-------: | :-------: | :-----------: | :-------: |
> > |             |       |                     |   R@10    |      R@20       |   R@50    |   R@10    |     R@20      |   R@50    |
> > |             |   1   |       STTran        |     -     |        -        |     -     |     -     |       -       |     -     |
> > |    Weak     |   2   | STTran$_{+Obj-PLA}$ |   14.32   |      20.42      |   25.43   |   14.78   |     21.72     |   30.87   |
> > |             |   3   |   STTran$_{+PLA}$   |   15.39   |      21.44      |   26.24   |   15.83   |     22.83     |   31.74   |
> > |             |   4   |  STTran$_{+GTRel}$  | **16.10** |    **22.38**    | **27.33** | **16.73** |   **24.02**   | **33.40** |
> >
> > Furthermore, the method to detect objects under unlocalized scene graph setting is not our main contribution. As a general framework, if better methods to detect objects are proposed, PLA can easily extend to them.
> >
> > ### **The generalization of PLA.**
> >
> > PLA is a model-agnostic method, which has two steps: first to assign pseudo labels for the video, then to train a fully-supervised VidSGG model by these pseudo labels. Therefore, PLA is agnostic to different VidSGG architectures, it can be easily incorporated into any advanced VidSGG model.
> >
> > We add another experiment about applying PLA to DSG-DETR[2], another VidSGG method. Results are shown in the Table below, we can observe that PLA with DSG-DETR (#3) also achieves similar performance improvements with STTran across all the metrics, e.g., in terms of R@10 with constraint criteria, #3 improves by 10.54% (15.42 v.s.13.95) relatively over #2, while PLA with STTran improves by 7.47% (15.39 v.s.14.32) relatively over baseline. These results show that PLA is a general framework.
> >
> > | Supervision |   #   |         Model         |         | With Constraint |         |         | No Constraint |         |
> > | :---------: | :---: | :-------------------: | :-----: | :-------------: | :-----: | :-----: | :-----------: | :-----: |
> > |             |       |                       |  R@10   |      R@20       |  R@50   |  R@10   |     R@20      |  R@50   |
> > |             |   1   |       DSG-DETR        |    -    |        -        |    -    |    -    |       -       |    -    |
> > |    Weak     |   2   | DSG-DETR$_{+Obj-PLA}$ |  13.95  |      20.08      |  25.10  |  14.29  |     21.25     |  31.08  |
> > |             |   3   |   DSG-DETR$_{+PLA}$   | $15.42$ |     $21.28$     | $25.89$ | $15.87$ |    $22.73$    | $31.86$ |
> >
> > **Corresponding Revision**: Sec. 4 (Line 283-316) in the revised paper. (We revised the response after ddl, so the content here is not exactly the same as that in the revised paper).
> >
> > [2] Feng S, Tripathi S, Mostafa H, et al. Exploiting Long-Term Dependencies for Generating Dynamic Scene Graphs. WACV2023.

---

> > > ### Author Response · Authors · 2022-12-01
> > > **Response to Reviewer spBp (3/3)**
> > >
> > >
> > > ## Q3: Relying on the image features instead video features.
> > >
> > > > This unfortunately remains an issue for the proposed method, as it relies on image features from an object detector (L285).
> > >
> > > **A3**: We mainly use the image features from the object detector for three main reasons:
> > >
> > > - One of our contributions is to propose a pseudo-label assignment method, it is model-agnostic. Improving the performance of the fully-supervised VidSGG method is not our contribution.
> > >
> > > - To show the effectiveness of PLA, we select one of the best models in frame-level VidSGG, for comparison. In that work, they only use image features but without video features. Following them, we also use image features for the proposed task.
> > >
> > > - Since PLA is model-agnostic, it can be used in any VidSGG method. If some more advanced VidSGG methods use video features, PLA can be applied to them and boost performance in the SF-VidSGG task.
> > >
> > > ## Q4: Typos.
> > >
> > > > I thought the baseline listed Table 2 is referring to model A in Table 1, yet the numbers do not quite match. ...... L322, I think the text is referring to Table 2 instead of Table 1.
> > >
> > > **A4**: Thanks for pointing out these typos. We have revised all the typos and further polished our presentation in the new manuscript.

---

> > > > ### Comment · Reviewer_spBp · 2022-12-13
> > > > **Thanks for the response**
> > > >
> > > > I appreciate the author response to my comments. Most of my prior concerns are now addressed. And I am happy to raise my rating.
> > > >
> > > > One remaining concern is related to Q1. While I accept that this paper presents "a new framework for a new task", I have to point out that the framework is a combination of existing techniques (see my previous comments), and the new task is an extension of an existing task (weakly supervised image scene graph generation). While I am fine with the main claim on the contributions, the paper looks less exciting for me.
> > > >
> > > > To strengthen the paper, it will be great to see some of the insights from the model design / results. For example, the new results on the upper bound performance is quite interesting. It seems to suggest that the main bottleneck for this problem lies in the pre-trained object detector. I would encourage the authors to reflect the design / results in the main paper.

---

> > > > > ### Author Response · Authors · 2022-12-13
> > > > > **Thanks**
> > > > >
> > > > > Thank you so much for raising your rating to positive, and we will clarify all the misunderstandings in the revision.

---

### Decision · Program_Chairs · 2023-01-20

**Decision:**

Accept: poster

**Justification For Why Not Higher Score:**

This paper presents a new task and also designs a new solution, however, there are still limitations in paper writing and experiments.

**Justification For Why Not Lower Score:**

This paper presents a new task and also designs a new solution, and reviewers have positive ratings.

**Metareview: Summary, Strengths And Weaknesses:**

This paper investigates a new task, i.e., weakly supervised video scene graph generation (VidSGG). The authors then proposed a novel approach to address this challenging task, and the key idea is to generate pseudo labels for unannotated frames. Experimental results on the Action Genome dataset are reported and discussed.

Overall, this paper is well written and clearly organized. A major contribution of this work is to present a new task, i.e., weakly supervised VidSGG. The proposed method is well motivated and easy to follow, and the experiments can demonstrate the effectiveness of the method.

Reviewers raised some concerns regarding novelty, technical details and experimental settings. Reviewers provided detailed responses to all questions as well as additional results, which addressed most of the concerns from reviewers. Consequently, reviewers raised the scores. The authors are strongly encouraged to incorporate the suggestions from reviewers in the final version.

**Note From Pc:**

if the above contains the word "oral" or "spotlight" please see: "oral" presentation means -> notable-top-5% and "spotlight" means -> notable-top-25%. As stated in our emails, we are disassociating presentation type from AC recommendations

**Summary Of Ac-Reviewer Meeting:**

N/A